# More Thinking, Less Seeing? Assessing Amplified Hallucination in Multimodal Reasoning Models

**Zhongxing Xu**[1,3*]    **Chengzhi Liu**[2*]    **Qingyue Wei**[1]    **Juncheng Wu**[3]
**James Zou**[1]    **Xin Eric Wang**[2]    **Yuyin Zhou**[3]    **Sheng Liu**[1]
[1] Stanford University    [2] UC Santa Barbara    [3] UC Santa Cruz

chengzhi@ucsb.edu, zhongxingxu1@gmail.com, shengl@stanford.edu

## Abstract

Test-time compute has empowered multimodal large language models to generate extended reasoning chains, yielding strong performance on tasks such as multimodal math reasoning. However, we observe that this improved reasoning ability often comes with increased hallucination: as generations become longer, models tend to drift away from image-grounded content and rely more on language priors. Attention analysis reveals that longer reasoning chains reduce focus on visual inputs, contributing to hallucination. To systematically study this phenomenon, we introduce *RH-AUC*, a metric that quantifies how a model's perception accuracy changes with reasoning length, enabling evaluation of whether the model preserves visual grounding while reasoning. We also release *RH-Bench*, a diagnostic benchmark covering diverse multimodal tasks, designed to jointly assess the balance of reasoning ability and hallucination. We find that *(i)* larger models generally exhibit a better balance between reasoning and perception; *(ii)* reasoning and perception balance depends more on the types and domains of the training data than its volume. Our findings highlight the need for evaluation frameworks that account for both reasoning quality and perceptual reliability.

## 1   Introduction

Large reasoning models scale test-time computation to improve complex reasoning. These models [6, 7, 32, 2] generate longer outputs and engage in deeper reasoning before producing final answers, resulting in more comprehensive solutions for complex mathematical and scientific problems. This paradigm has been extended to multimodal large language models: non-reasoning base models are supervised finetuned (SFT), or finetuned with reinforcement learning (RL) to obtain strong reasoning ability [36, 56, 10, 49, 54, 21], demonstrating exceptional capabilities in multimodal reasoning tasks, particularly in domains like mathematical problem solving.

Most existing studies on multimodal reasoning models focus on enhancing reasoning performance, with limited attention paid to perception-focused tasks. As illustrated in Figure 1a, although the reasoning model generates an extended reasoning chain in visual question answering, its answer is largely driven by language priors rather than visual evidence, leading to hallucination. Our empirical study reveals a consistent and significant finding: although reasoning models can generate more detailed reasoning chains, they introduce more hallucinations in perception-focused tasks than the non-reasoning counterparts, as shown in Figure 1b.

Through attention analysis, we investigate the decrease of attention on visual tokens in multimodal reasoning models, which exacerbates visual hallucinations. The reasoning model allocates significantly less attention to visual tokens compared to its non-reasoning counterpart, while directing more attention to the instruction tokens. This bias increases reliance on language priors and amplifies

---

[1] `https://mlrm-halu.github.io/`. Work was partially done while ZX was visiting Stanford.

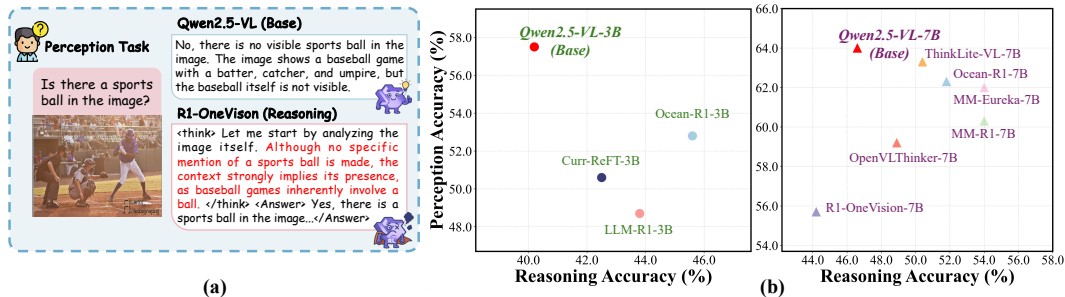

Figure 1: **(a)** Example of outputs from a reasoning model and a non-reasoning model on a perception task. Red highlights indicate visual hallucination. *Multimodal reasoning models are generally more prone to amplifying hallucinations during the reasoning process compared to their non-reasoning counterparts.* **(b)** Performance of different models on reasoning and perception tasks in the *RH-Bench* dataset. Better performing models are positioned in the upper right corner. *Baseline non-reasoning models of varying scales typically exhibit weaker reasoning capabilities and fewer hallucination, whereas reasoning models display the opposite trend.*

hallucination risk. Moreover, the extension of the reasoning chain further weakens the visual attention allocation, leading to an increase in hallucinations, as the model becomes more dependent on language-based reasoning rather than visual evidence.

Based on these findings, we further investigate the impact of reasoning chain length on model reasoning and hallucination. The results indicate that the influence of reasoning chain length on reasoning-hallucination exhibits a non-monotonic relationship. Additionally, the optimal reasoning range differs across tasks, while traditional evaluation metrics, such as accuracy and hallucination rate, are inadequate for capturing the dynamic balance between reasoning and visual grounding.

To address this, we introduce *RH-AUC*, a new metric designed to assess the balance between reasoning and hallucination in multimodal reasoning models. This metric is computed by calculating the area under the curve formed by reasoning performance and hallucination performance at different reasoning lengths, with higher values indicating better balance. Alongside this metric, we release *RH-Bench*, a diagnostic benchmark containing 1,000 samples across various reasoning and perception tasks, with each task featuring both multiple-choice questions and open-ended questions. Through the evaluation of *RH-Bench*, we observe three key findings: *(i)* Larger models typically demonstrate better reasoning and hallucination balance. *(ii)* RL-only training models promote more adaptive reasoning, resulting in a better balance between reasoning and hallucination compared to SFT+RL. *(iii)* Reasoning-Hallucination balance is more influenced by the types and domains of the training data than by its volume. To sum up, our contributions are listed as follows:

- We observe that multimodal reasoning models are more prone to hallucinations than their non-reasoning counterparts in perception tasks, which can be attributed to a decline in visual attention allocation. Longer reasoning chains further diminish visual attention.
- We reveal that the relationship between reasoning chain length and the model's reasoning and perception performance is non-monotonic, with the optimal length varying across tasks.
- We introduce the new *RH-AUC* metric and the *RH-Bench* diagnostic dataset to systematically evaluate the balance between reasoning and hallucination across varying reasoning lengths in multimodal reasoning models.

## 2 Multimodal Reasoning Can Amplify Visual Hallucination

In this section, we begin by investigating whether multimodal reasoning models introduce more hallucination in perception-focused tasks. Specifically, we compare 8 recent multimodal reasoning models against their backbone non-reasoning-based counterparts across multiple hallucination benchmarks, including MMVP [42], MMEval-Pro [13],VMCBench [61],Bingo [5],MMHAL [39].

### 2.1 Hallucination Increases Consistently Compared to Base Models

To systematically assess the impact of multimodal reasoning on visual grounding, we evaluated eight reasoning-augmented models against their non-reasoning Qwen2.5-VL backbones on five

hallucination datasets. As shown in Figure 2, all reasoning models trace markedly smaller radar areas than their baselines, indicating uniformly higher hallucination rates on perception-focused tasks. This deficit remains consistent at both the 3 B and 7 B scales, demonstrating that the elevated hallucination rate stems from the reasoning paradigm itself rather than model size.

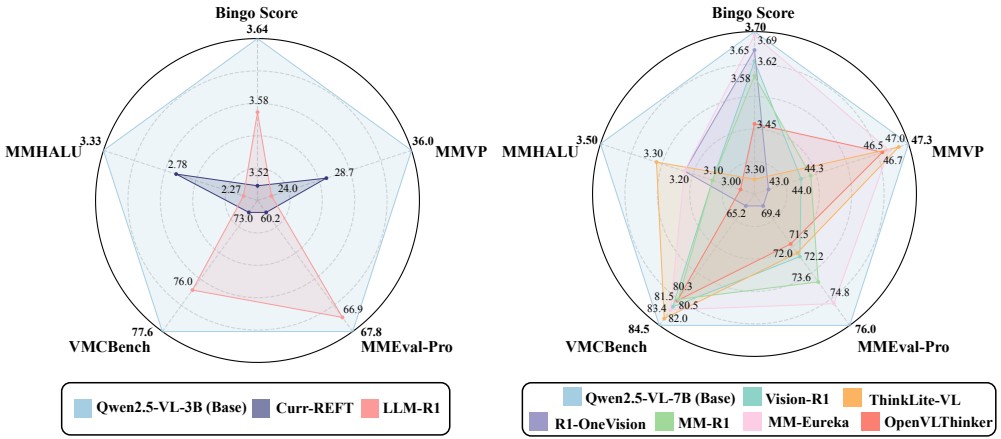

Figure 2: Comparison of reasoning and non-reasoning models on five perception benchmarks. Results are shown for 3B models (left) and 7B models (right). Higher scores indicate lower hallucination.

## 2.2 Does Training Paradigm Matter? Comparison Between RL and SFT+RL

Current multimodal reasoning models typically adopt one of two training regimes: (1) pure reinforcement learning (RL-only) or (2) supervised fine-tuning followed by reinforcement learning (SFT+RL). Figure 3 shows a consistent performance hierarchy across four perception benchmarks: The Qwen2.5-VL baseline achieves the highest scores, followed by RL-only fine-tuning, with the SFT+RL pipeline performing the worst. This pattern highlights the robustness of baseline model in visual grounding and indicates that subsequent RL or hybrid fine-tuning weakens this robustness, with the supervised-preceded RL strategy leading to the most significant performance degradation.

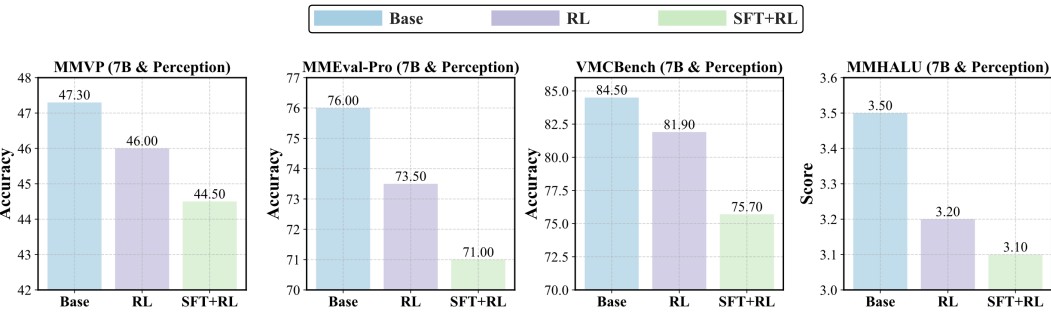

Figure 3: Performance across four perception benchmarks comparing Base, RL, and SFT+RL.

## 2.3 Case Study of Hallucinations in Reasoning Models

Figure 4 presents two representative hallucination patterns observed in multimodal reasoning models, arising from visual misrecognition and reasoning bias, respectively. In Figure 4a, the reasoning model fails to identify fine-grained visual cues and miscounts four individuals as three, reflecting a localized deficiency in visual perception. In Figure 4b, the reasoning model increasingly relies on linguistic priors during the reasoning process while overlooking early visual evidence, ultimately generating an incorrect response. In contrast, the baseline model exhibits a lower hallucination rate under identical inputs. These observations raise a crucial question: why do multimodal reasoning models, despite their strong reasoning performance, exhibit weakened visual grounding? In the next section, we provides an in-depth analysis based on the internal attention mechanisms of the reasoning models.

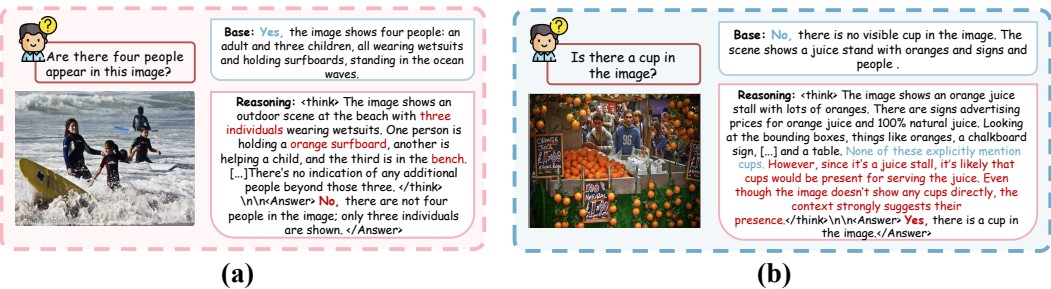

**(a)**           **(b)**

Figure 4: Two common types of hallucination patterns observed in multimodal reasoning models. (a) corresponds to hallucinations caused by visual misrecognition, while (b) reflects hallucinations arising from reasoning biases. Hallucinated spans are highlighted in red.

> **Takeaway 1: Reasoning Models Amplify Visual Hallucinations**
>
> *Across training paradigms and model scales, multi-modal reasoning models exhibit a consistent drop in accuracy and rise in hallucination rates on general visual benchmarks.*

## 3 Why Reasoning Models Amplify Hallucinations?

Many previous studies have investigated the role of attention mechanisms in hallucination, identifying insufficient attention allocation as a potential key factor contributing to hallucinations[14, 16, 53]. In this section, we conduct an attention based analysis to explore the underlying causes of hallucination amplification in multimodal reasoning models. Section 3.1 indicates that hallucinations may result from limited attention allocated to visual inputs, while Section 3.2 shows that longer reasoning chains further weaken the model's visual focus.

### 3.1 Hallucination Resulting from Weak Visual Attention

We conduct a comparative analysis of the attention distributions over visual, instruction, and system tokens across all layers in the reasoning and non-reasoning models. As shown in Figure 5a, the reasoning model consistently assigns low attention to visual tokens, with a further decrease observed in deeper layers, indicating a limited ability to integrate visual evidence. Meanwhile, more attention is shifted to instruction tokens, reflecting a heightened reliance on linguistic priors. In contrast, the non-reasoning maintains a relatively high and stable level of visual attention from shallow to intermediate layers. The visual attention heatmap in Figure 5b further supports this observation: while the non-reasoning model progressively focuses on semantically salient regions, the reasoning model exhibits sparse and dispersed attention, failing to consistently engage with key visual areas. This phenomenon indicates that the weakening of visual attention undermines the reasoning model's ability to achieve effective visual grounding, exacerbating the occurrence of hallucinations.

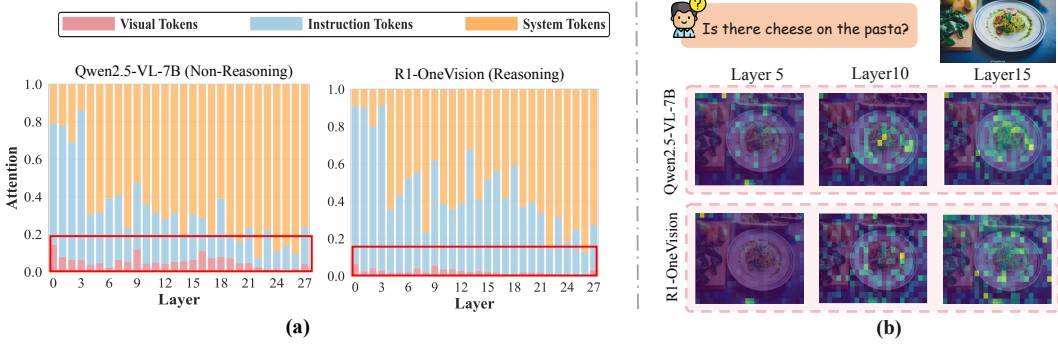

Figure 5: Attention allocation and visual grounding between reasoning and non reasoning models. *The reduction of visual attention in reasoning models amplifies visual hallucinations.*

## 3.2 Visual Focus Declines with Longer Reasoning Chains

As shown in Figure 6, we visualize the attention distributions of the reasoning model under two reasoning modes: normal thinking and overthinking. As the reasoning chain length increases, the heatmaps clearly reveal a systematic shift in the model's attention focus: under the overthinking mode, attention to visual tokens significantly decreases, while attention to instruction tokens intensifies. This pattern indicates that longer reasoning chains cause the model to increasingly rely on linguistic cues rather than grounded visual evidence. For instance, when asked whether a gray wall is present, the model under normal thinking correctly identifies the gray well and provides a correct response. In contrast, under over-reasoning conditions, the model exhibits further diminished attention to visual tokens, with increased focus directed toward the end of the user instruction. This suggests that longer reasoning chains tend to further exacerbate the degradation of the model's visual grounding, potentially leading to an increase in hallucinations.

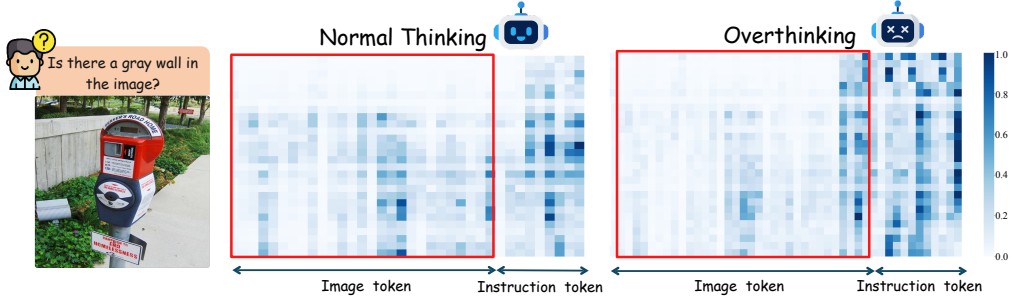

Figure 6: Attention shift in the reasoning model under different reasoning length. In normal thinking, the model generates outputs as typically expected, while in overthinking, the reasoning length is adjusted using Latent State Steering (Section 4.1). *Longer reasoning chains further exacerbate the degradation of attention to visual information and focus toward linguistic priors.*

## 4 Effects of Reasoning Length on Reasoning-Hallucination Balance

In this section, we explore the impact of reasoning length on the balance between hallucination and reasoning. Section 4.1 provides an overview of the proposed control strategy: latent state steering as well as techniques that are previously used in the literature[31]: budget forcing, and test time scaling. In Section 4.2, we explore the optimal generation length for various benchmarks and analyze the trade-off between hallucination and reasoning performance as reasoning length varies.

### 4.1 Overview of Reasoning Length Control Straregies

To systematically control the reasoning length in reasoning models, we adopt three strategies:

*(1) Token Budget Forcing*: A hard constraint on reasoning length is enforced by predefining a generation budget at decoding time, directly limiting the number of tokens allocated for the reasoning.

*(2) Test Time Scaling*: Reasoning is incrementally extended during inference through staged generation. The model first produces partial reasoning under a 4096-token constraint and halts midway. It is then prompted to continue by appending a simple token ("Wait"), enabling soft extension of reasoning while preserving contextual coherence.

*(3) Latent State Steering*: Inspired by recent works on latent space steering for behavior control in large language models [23, 22, 1, 29], we propose a method to steer the model toward generating reasoning traces of varying lengths. Specifically, we extract steering directions from the post-attention hidden states by computing the difference of latent states between long and short reasoning trajectories. These direction vectors are obtained and applied across all layers of the text decoder, with a scaling factor controlling both the magnitude of guidance on the reasoning length. Specifically, we collect responses from the test benchmark and categorize them into long reasoning traces $\mathcal{R}_{\text{long}}$ and short reasoning traces $\mathcal{R}_{\text{short}}$ based on token length. The query and reasoning steps for each sample are input into the model, from which hidden representations $S^\ell$ are extracted at each layer. $S^\ell(q, t)$ denotes the hidden representation at layer $\ell$ for token position $t$ in the response to query $q$. We

compute the average hidden representation over reasoning tokens, where $\mathcal{H}_i$ represents the set of token positions within the reasoning span. The average representation is then calculated across the long and short reasoning traces to obtain layerwise embeddings:

$$\mathcal{S}_{\text{long}}^{\ell} = \frac{1}{|\mathcal{R}_{\text{long}}|} \sum_{q \in \mathcal{R}_{\text{long}}} \frac{1}{|\mathcal{H}_i|} \sum_{t \in \mathcal{H}_i} S^{\ell}(q, t), \quad \mathcal{S}_{\text{short}}^{\ell} = \frac{1}{|\mathcal{R}_{\text{short}}|} \sum_{q \in \mathcal{R}_{\text{short}}} \frac{1}{|\mathcal{H}_i|} \sum_{t \in \mathcal{H}_i} S^{\ell}(q, t) \quad (1)$$

The reasoning length direction at layer $\ell$ is defined as the difference between the long and short embeddings, denoted as $d^{\ell}$, which captures the variation in the model's representation resulting from different reasoning chain lengths. To adjust the hidden representation based on this direction, We introduce a parameter $\alpha \in [-0.15, 0.15]$ to dynamically control the reasoning length and its magnitude. As $\alpha$ increases, the length of the reasoning chain extends, as shown below:

$$d^{\ell} = \mathcal{S}_{\text{long}}^{\ell} - \mathcal{S}_{\text{short}}^{\ell}, \quad \mathcal{S}_{\text{steering}}^{\ell} = \mathcal{S}^{\ell} + \alpha d^{\ell}. \quad (2)$$

These strategies are applied to five representative multimodal reasoning models and evaluated on six benchmark datasets, covering both reasoning and perception tasks. In Figure 7, we present two benchmarks for both tasks. All implementation details and results are provided in Appendix C.

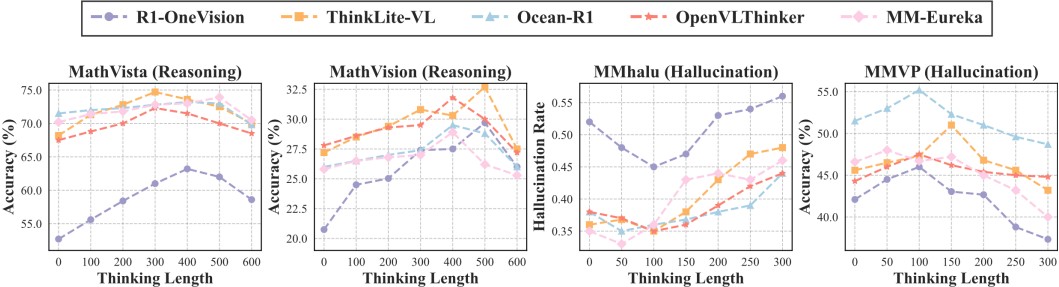

Figure 7: Reasoning-Hallucination balance of multimodal reasoning models under varying reasoning lengths. Thinking lengths are controlled within [0–600] tokens for reasoning and [0–300] for hallucination, corresponding to the longer chains required for reasoning and shorter for hallucination.

## 4.2 Dynamic Balance Between Reasoning and Hallucination

**Non-monotonic Effect of Reasoning Length on Reasoning and Perception Performance.** The relationship between reasoning length and model performance typically exhibits a non-monotonic pattern under reasoning and perception tasks. Across various length control strategies, a consistent trend emerges: *moderate reasoning depth tends to yield optimal performance, whereas overly short or excessively long reasoning chains often lead to a decline in accuracy.* As shown in Figure 7, we employ the *Latent State Steering* method adjusts the thinking step for reasoning and perception tasks. It is evident that as the thinking length increases, the model's performance across tasks generally follows a rising-then-falling trajectory. This indicates that enhanced reasoning does not linearly improve model performance, but instead follows a dynamic trade-off pattern.

**Task-Specific Variability of Optimal Reasoning Intervals.** While most tasks exhibit non-monotonic relationships between reasoning length and performance, we further observe that the optimal reasoning length varies significantly across tasks. Figure 7 reveals that reasoning benchmarks such as MathVista [27] tend to benefit from longer reasoning chains, whereas perception and hallucination-oriented tasks such as MMHalu achieve their best performance at shorter or moderate lengths. This indicates that the balance between reasoning depth and performance is task-specific, and unified length control strategies are unlikely to be effective across all task types.

**Impact of the Zerothink Condition.** Zerothink retains the reasoning structure but lacks substantive content. As shown in Figure 7, this setting leads to a consistent drop in model performance on both reasoning and perception benchmarks, notably lower than results under normal reasoning lengths. These results indicate that the absence of reasoning content diminishes the reasoning model's performance in both perception and reasoning.

**Limitation of Conventional Metric.** Conventional metrics like reasoning accuracy and hallucination rate, when computed at a fixed generation length, fail to capture the dynamic balance between deeper

reasoning and perception. Figure 7 shows that reasoning and perception often peak at different reasoning lengths, making it misleading to evaluate models using single-point metrics or simple averages between reasoning and hallucination performance. For instance, a short reasoning trace may yield a lower hallucination rate but poor reasoning depth, while a longer trace may improve reasoning at the cost of increased hallucination, yet both scenarios could yield the same average score.

To capture this evolving balance, in the next section, we propose an AUC-style metric that summarizes the balance curve between reasoning and perception fidelity across various reasoning lengths. This provides a more faithful and holistic measure of performance, revealing both the model's optimal balance and its stability across varying generation lengths.

> **Takeaway 2: Moderate Reasoning Length Strikes the Best Reasoning-Hallucination Balance**
>
> *Reasoning length exerts a non-monotonic effect on model performance: both insufficient and excessive reasoning degrade accuracy, and the optimal length is task-dependent.*

## 5 Evaluation on the Reasoning-Hallucination Balance

To comprehensively quantify the balance between reasoning and hallucination in multimodal large reasoning models at different reasoning depths, we introduce a new metric **RH-AUC**. This metric captures how hallucination risk evolves with reasoning depth while also reflecting the cumulative effects of reasoning and perception. Additionally, we present **RH-Bench**, a new diagnostic dataset of 1000 samples, designed for the integrated evaluation of reasoning and perception tasks, offering a robust basis for analyzing reasoning ability and perceptual hallucinations.

### 5.1 Setup

**Benchmark Overview.** *RH-Bench* consists of two types of tasks: reasoning and perception, with each task including two types of questions: multiple-choice and open-ended. The reasoning task includes 500 samples sourced from MathVision [44], MathVista [27], MMMU [55], and ScienceQA [28], while the visual perception task includes 500 samples from MMhalu, MMVP, HallusionBench, and VMCBench. Both task types use accuracy as the evaluation metric. For multiple-choice questions, evaluation is based on matching the final options. For open-ended questions, both tasks are evaluated using GPT-4o. The reasoning task determines whether the generated response is consistent with the correct answer, whereas the visual task evaluates the generated response against the correct answer, assigning a score within the range of 0 to 6. Responses with a score below 3 are classified as hallucinations. All sample ground-truth and evaluation answers have undergone manual inspection.

| Method | Paradigms | Perception | | Reasoning | | Training Data | | RH-AUC |
|---|---|---|---|---|---|---|---|---|
| | | Acc.(%) ↑ | Length | Acc.(%) ↑ | Length | Perc. | Reas. | |
| LLM-R1-3B | RL | 48.7 | 121.9 | 43.8 | 391.8 | 65k | 40k | 0.46 |
| Curr-ReFT-3B | SFT+RL | 50.6 | 133.7 | 42.5 | 472.61 | 6k | 3k | 0.47 |
| Ocean-R1-3B | RL | 52.8 | 131.2 | 45.6 | 414.5 | 20k | 63k | 0.53 |
| R1-OneVision-7B | SFT+RL | 55.7 | 162.9 | 44.2 | 457.3 | 80k | 77k | 0.46 |
| ThinkLite-VL-7B | RL | 63.3 | 110.4 | 50.4 | 435.4 | 62k | 8k | 0.52 |
| OpenVLThinker-7B | SFT+RL | 59.2 | 187.7 | 48.9 | 460.1 | 25k | 25k | 0.54 |
| MM-Eureka-7B | RL | 62.0 | 139.6 | 54.0 | 450.5 | - | 15k | 0.55 |
| MM-R1-7B | RL | 60.3 | 139.6 | 54.0 | 430.0 | - | 6k | 0.57 |
| Ocean-R1-7B | RL | 62.3 | 90.4 | 51.8 | 262.2 | 20k | 63k | **0.63** |

Table 1: Comparison of model performance on *RH-Bench*, including task-specific accuracy and *RH-AUC* scores. **Perc.** and **Reas.** respectively denote training data for visual perception and reasoning.

**RH-AUC** We define reasoning length as $T$, which controls the extent of the model's generated reasoning trace. For each length $T$, we compute $R_T$, which represents the reasoning performance at length $T$, and $H_T$, representing performance on hallucination at the same length.

By evaluating the model at multiple lengths on the *RH-bench* benchmark, we obtain a series of $(R_T, H_T)$ pairs that form a balance curve between reasoning and perceptual hallucination. To compute the area under this curve, we first sort the pairs in ascending order of reasoning performance $R_T$. Let

the sorted indices be denoted as $T^{(0)}, T^{(1)}, \ldots, T^{(n-1)}$, such that $R_{T^{(0)}} \leq R_{T^{(1)}} \leq \cdots \leq R_{T^{(n-1)}}$. To ensure comparability across models, both $R_T$ and $H_T$ are min-max normalized to the range $[0, 1]$. The **RH-AUC** is then computed using the trapezoidal rule as:

$$RH\text{-}AUC = \sum_{i=0}^{n-2} \frac{R_{T^{(i+1)}} - R_{T^{(i)}}}{2} \cdot \left(H_{T^{(i+1)}} + H_{T^{(i)}}\right), \tag{3}$$

where $n$ is the number of evaluated reasoning lengths. A higher **RH-AUC** indicates a model that better balances reasoning and hallucination across different reasoning lengths.

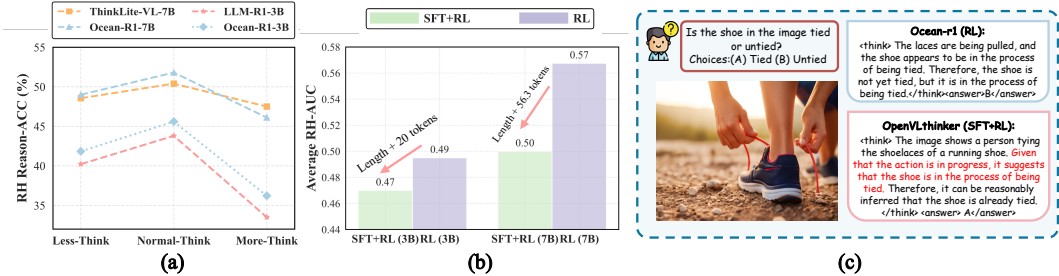

Figure 8: **(a)** Accuracy trends on the *RH-Bench* reasoning task across different reasoning lengths for 3B and 7B models. *Larger models typically exhibit more stable performance across varying reasoning lengths.* **(b)** Comparison of SFT+RL and RL-only training paradigms in terms of *RH-AUC*, with arrow directions indicating the increase in reasoning length for SFT+RL relative to RL-only. *RL-only training tends to generate more concise reasoning chains, leading to a better perception hallucination balance.* **(c)** Case study comparing RL-only and SFT+RL models. *SFT+RL models often introduce rigid imitation reasoning paths, which limit the flexibility of visual reasoning.*

## 5.2 Result Diagnosis

We conduct an in-depth analysis of model performance based on the evaluation results from the *RH-Bench* diagnostic dataset, investigating the influence of three key factors: model scale, training paradigm, and training dataset on the reasoning-hallucination balance.

**Model Scaling.** As shown in Table 1, the 7B model generally outperforms the 3B model in *RH-Bench*, demonstrating higher *RH-AUC*, primarily due to its larger parameter size and stronger representational capacity. As illustrated in Figure 8a, the larger model maintains higher stability, especially under longer reasoning chains, while the smaller models show a noticeable decline in performance. ***This suggests that larger models typically exhibit better robustness and adaptability.***

**Training Paradigms.** A comparison between the two-stage SFT+RL-trained model and the RL-only trained model reveals that RL demonstrates a stronger balance between reasoning and perception. For example, in Figure 8c, although the OpenVLThinker model maintains a longer reasoning chain, the introduction of redundant reasoning interferes with visual perception, leading to an incorrect inference that the shoe is already tied. In contrast, the RL-only model, Ocean-R1, uses a shorter reasoning chain, enabling it to more efficiently capture key visual features and avoid unnecessary complex reasoning steps. This advantage is particularly evident at different reasoning length, as shown in the average *RH-AUC* in Figure 8b, which is significantly higher for the RL-only model compared to SFT+RL. This phenomenon suggests that ***although SFT helps the model learn reasoning formats, it may introduce rigid imitation reasoning paths, limiting the model's adaptability to dynamic tasks and ultimately resulting in redundant reasoning.*** In contrast, RL encourages the model to generate more adaptive reasoning behaviors, enhancing the integration of reasoning and perception.

**Training Dataset.** The diversity and quality of training data play a crucial role in the reasoning-hallucination balance of models. Through a statistical analysis of the multimodel reasoning models training data and a comparison with the results in Table 1, we have observed some interesting phenomena: ***(1) More visual perception data does not necessarily improve the balance between reasoning and perception.*** Increasing the training samples of visual perception data can enhance the balance of reasoning models to a certain degree. For example, the ThinkLite-VL model, supported by large scale visual perception data, demonstrates strong hallucination and reasoning balance. Similarly,

the Ocean-R1 model adopts a two-stage training strategy, first enhancing reasoning ability and then strengthening visual perception, achieving the highest *RH-AUC* on *RH-bench*. However, this phenomenon is not consistent. For example, despite the R1-OneVision model utilizing a large amount of visual perception data, it demonstrates a weaker balance between reasoning and perception, which may be attributed to the limitations of its training paradigm design. *(2) Perception and Reasoning balance can be achieved through training on domain-specific data.* Training on domain-specific data helps enhance the balance of the reasoning model. For example, the MM-Eureka model, trained on a larger mathematical dataset, shows a higher *RH-AUC*, proving its effectiveness in balancing reasoning and perception. Similarly, despite being trained on only 6k mathematical data, the MM-R1 model still performs well on *RH-bench*. This highlights the potential of domain-specific data to stimulate the balance capabilities of reasoning models, even with smaller datasets. *(3) The size of the training data is not always a guarantee for the reasoning-perception balance.* The traing data size does not always directly correlate with the model's balance capability. For example, both the LLM-R1, trained on over 60k visual perception samples, and the R1-OneVision, with a dataset of 150k samples, exhibit inadequate reasoning-hallucination balance, with the *RH-AUC* of only 0.46.

# 6    Related Work

**Multimodal Reasoning Tasks.**  Multimodal reasoning requires integrating information across modalities to solve complex problems. It is generally categorized into general reasoning and domain-specific reasoning. General reasoning typically occurs in natural image scenarios, where models must combine visual perception with knowledge and commonsense. Representative benchmarks include multiple-choice datasets such as MMMU [55], MMVP [42], MMBench [24], MMStar [3], MMEval-Pro [13], and VMCBench [61], as well as open-ended evaluations like Bingo [5], MMHAL-Bench [13], POPE [19], CHAIR [35], and HallusionBench [12]. Domain-specific reasoning focuses on technical tasks within particular domains. For mathematical reasoning, benchmarks such as MathVista [27], MATH-Vision [44], MM-Math [37], WeMath [34] evaluate models' ability to solve math problems grounded in visual contexts. For physical reasoning, datasets like PhysBench [4] and CRAVE [38] test understanding of physics and commonsense reasoning from visual inputs.

**Reinforcement Learning in MLLMs.** Recent approaches enhance the reasoning capabilities of multimodal large models by incorporating chain-of-thought supervision during supervised fine-tuning or reinforcement learning [62, 58, 48, 41, 50, 52]. Methods like RLHF-V [54], LLaVA-Reasoner [60], and Insight-V [11] leverage large-scale CoT-style datasets and preference optimization to improve model reasoning. Following DeepSeek-R1, the GRPO (Group Relative Policy Optimization) algorithm has become a standard paradigm in training multimodal large reasoning models [26, 57, 47, 43, 18, 45, 25]. Some models, such as R1-OneVision [51], Reason-RFT [40], and R1-VL [59], follow a two-stage SFT + RL pipeline, while others like Ocean-R1 [20], ThinkLite-VL [46], and MM-Eureka [30] apply rule-based reinforcement learning directly at scale.

# 7    Conclusion

In conclusion, this paper investigates the balance between reasoning and hallucination in multimodal reasoning models, with a focus on how reasoning chain length and visual attention allocation impact performance. While longer reasoning chains enhance performance on complex tasks, they also exacerbate hallucinations by diminishing visual attention and increasing reliance on language priors. To address these challenges, the paper introduces the *RH-AUC* metric and the *RH-Bench* benchmark, which provide a systematic method to evaluate the balance between reasoning ability and hallucination risk. The findings reveal that reasoning-augmented models are more prone to hallucinations, highlighting the importance of developing evaluation frameworks that assess both the quality of reasoning and the accuracy of perception.

**Limitation.** Although our study provides a comprehensive analysis of visual hallucinations in multimodal reasoning models, it also has several limitations. First, our evaluation is limited to models built on the Qwen2.5-VL backbone, which may constrain the generalizability of our findings to architectures with different modalities or pretraining objectives. Second, our analysis of the influence of training data is based solely on technical reports and publicly available documentation of existing models, without conducting controlled retraining experiments. Therefore, our conclusions are observational and may not fully capture causal effects.

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

# Appendix

## A    Detailed Experimental Settings

**Datasets.** We evaluate our model on both discriminative and generative datasets, as listed below. (a) MMVP [42] evaluates recognition and reasoning performance across nine categories of basic visual patterns. (b) MMEval-Pro [13] assesses cross-modal understanding through triplet-based object and attribute recognition in natural images. (c) VMCBench [61] use adversarial distractors to test fine-grained discriminative ability across diverse tasks such as commonsense reasoning, image-text matching. All three datasets (a–c) adopt accuracy as the evaluation metric. (d) Bingo [5] evaluates bias and interference hallucinations, with GPT-4o used to score hallucination severity and response quality. (e) MMHAL-Bench [39] evaluates model capabilities beyond object hallucination, with GPT-4o used to assess hallucination rate and response informativeness.

**Implementation Details.** We select nine representative multimodal reasoning models to evaluate their hallucination performance on general vision tasks. We categorize these models into two major training paradigms: (1) the RL-only paradigm, where models are trained solely via reinforcement learning, including LMM-R1 [33], MM-R1[17], ThinkLite-VL[46], MM-Eureka[30], and Ocean-R1[20]; (2) the two-stage paradigm, combining supervised fine-tuning (SFT) with reinforcement learning, including Vision-R1[15], R1-OneVision[51], OpenVLThinker[9], and Curr-ReFT [8]. All models are post-trained on Qwen2.5-VL-3B or Qwen2.5-VL-7B, which are used as baseline models.

## B    Reasoning Models Attention-Based Analysis

### B.1    Visual Attention Heatmap

Figure 9-10 compares the visual attention distribution between multimodal reasoning models and their corresponding non-reasoning models. The results indicate that, compared to non-reasoning models, reasoning models exhibit weaker focus on key image details, with attention more dispersed across other regions of the image. Specifically, reasoning models display a greater degree of attention dispersion at lower layers, and their attention is not concentrated on critical areas of the image. In contrast, non-reasoning models demonstrate more precise visual grounding. For instance, as shown in Figure 10, the attention maps of the non-reasoning model at layers 10 and 15 consistently focus on the target object, the white mouse, highlighting its sustained attention on the target.

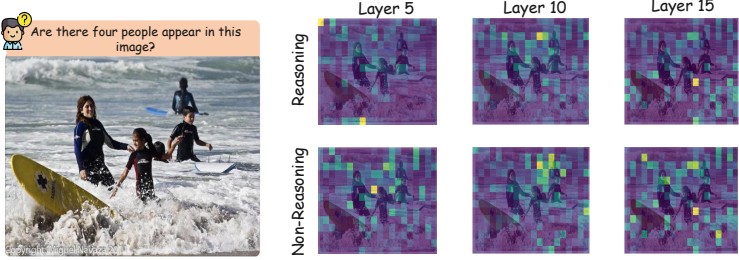

Figure 9: Case Study 1: Attention Heatmap in Counting Tasks.

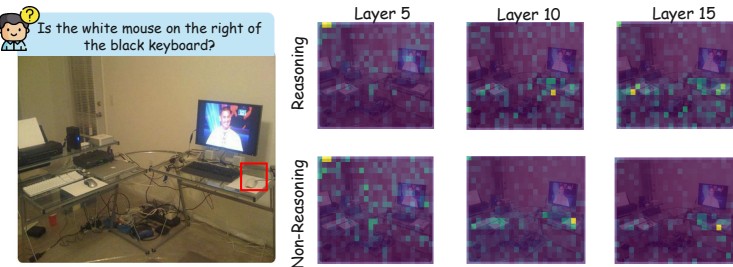

Figure 10: Case Study 2: Attention Heatmap in Object Localization.

## B.2 More Examples of the Impact of Reasoning Length on Visual Perception Degradation

This section presents an additional example, including a visual task involving counting, comparing the results of reasoning models and their corresponding attention maps under different reasoning lengths. It is evident that an excessively lengthy reasoning process causes the model to disregard the visual information inherent in the image, instead relying more heavily on prior linguistic knowledge. In Figure 11, the attention maps clearly show that, under over reasoning conditions, the model's attention shifts more towards the instruction tokens following the image tokens, particularly towards the latter part of the instruction. This suggests that prolonged reasoning reduces the model's focus on the visual information, leading it to depend more on the guidance provided by the linguistic instructions.

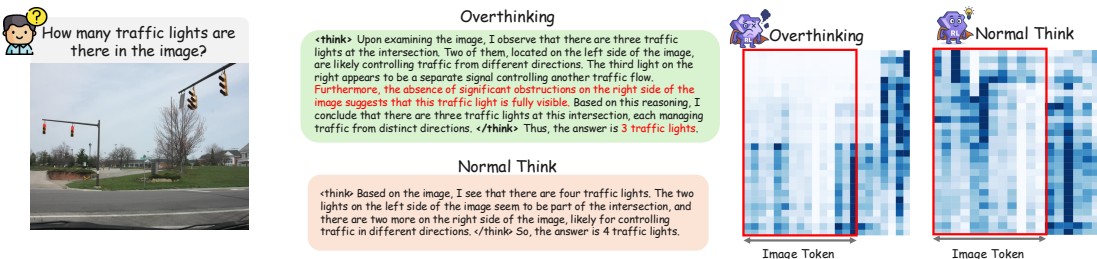

Figure 11: Attention shift in the reasoning model under different reasoning length.

## B.3 Error Analysis

In this section, we further focus on the error rates of multimodal reasoning models and non-reasoning models across different problem types, conducting a statistical analysis to compare the differences between the two. Figure 12a presents the error type statistics for the Bingo benchmark samples. By combining GPT-4o evaluations with manual inspection, we analyze the reasoning process and final answers of the reasoning model to determine whether the errors stem from reasoning or perception. If the model's errors arise from both reasoning and perception, we classify them as "perception and reasoning" errors. The statistical results indicate that the proportion of perception errors in the reasoning model decreases, with more errors originating from the reasoning process. This suggests that the reasoning model does not completely fail to interpret the image information, but rather diminishes its focus on perceptual information during reasoning. The evaluation results in Figure 12b further confirm this phenomenon: the overall error rate of the reasoning model is higher than that of the non-reasoning model, with a more prominent proportion of errors coming from reasoning.

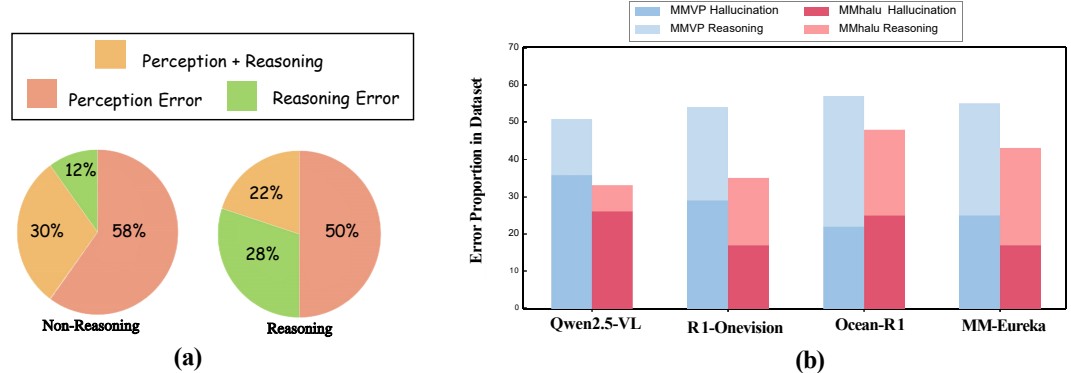

Figure 12: Error type distribution and error proportions across reasoning and non-reasoning models in Bingo benchmark. (a) Pie charts showing the distribution of perception and reasoning errors for non-reasoning and reasoning models, with the breakdown of perception error, reasoning error, and combined perception & reasoning errors. (b) Bar chart illustrating error proportions in the MMVP and MMhalu benchmarks, comparing hallucination and reasoning errors across reasoning models.

# C Reasoning Length Control

## C.1 Comparison of Three Reasoning Length Control Straregies

In the manuscript, we have thoroughly explored three methods: *Token Budget Forcing*, *Test Time Scaling*, and *Latent State Steering*. The first two methods directly control the model's reasoning length by using fixed-length truncation or soft expansion of the reasoning length, ensuring dynamic expansion within a predefined thinking length range. However, the limitations of *Token Budget Forcing* and *Test Time Scaling* are that they can only control the model's reasoning length to shorten or lengthen, lacking flexibility for more nuanced adjustments. In contrast, Latent State Steering introduces a tuning coefficient $\alpha$, allowing more flexible control over the model's reasoning length. By adjusting the value of $\alpha$, we can effectively quantify changes in the extent of reasoning. All of our Latent State Steering experiments are dynamically adjusted within the range of $\alpha \in [-0.15, 0.15]$. Furthermore, in the subsequent *RH-bench* calculation of *RH-AUC*, it is precisely due to the flexibility of the Latent State *Latent State Steering Strategy* that we apply it to dynamically regulate the reasoning length and perform further quantification.

## C.2 Model Performance Variation of the *Latent State Steering* Length Control Strategy

Figures 13 presents the visualization of performance variations for different models under the *Latent State Steering* strategy, with $\alpha$ ranging from $[-0.15, 0.15]$. In Figure 13, the star symbol represents the performance under the base condition. It is clearly observed that the variation in reasoning length shows that the optimal intervals for reasoning models differ between reasoning and hallucination tasks, with both exhibiting non-monotonicity.

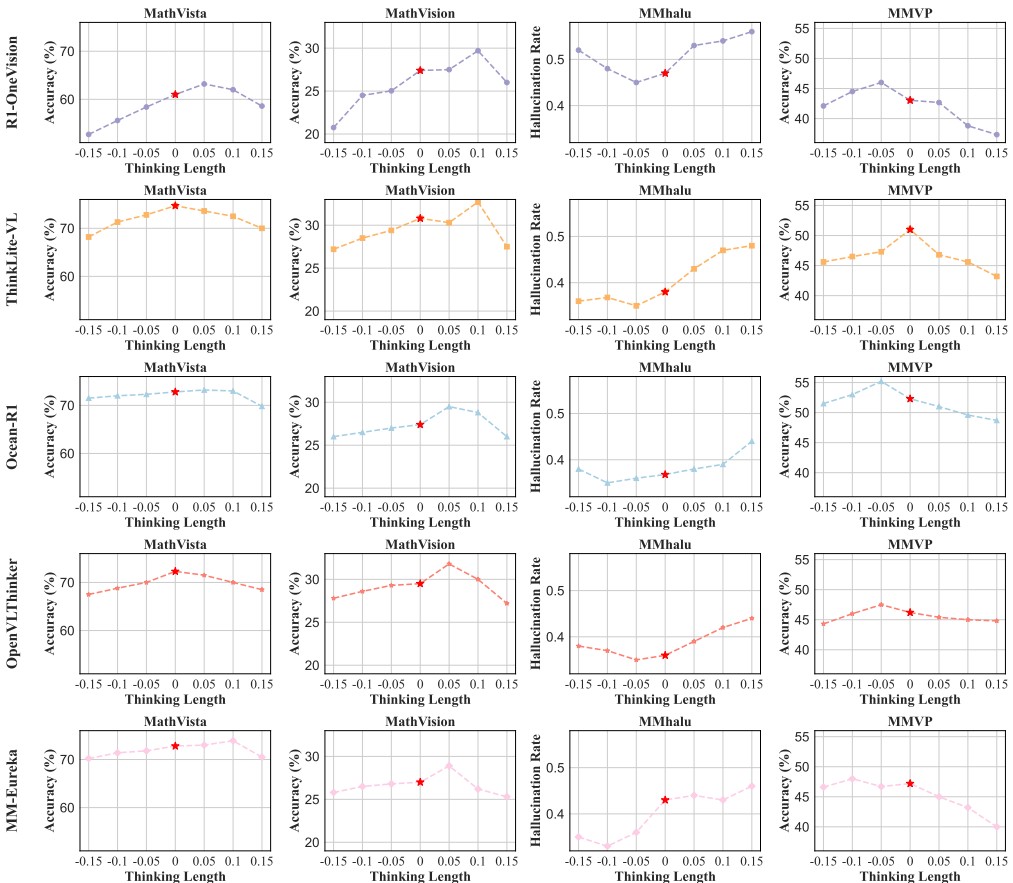

Figure 13: Model performance variation of the *Latent State Steering* strategy. The star symbol represents the original thinking length of the model without steering or test-time intervention.

# D  More Examples from *RH-Bench*

In this section, we present samples from different tasks and question types in the *RH-Bench* benchmark. As shown in Figures 14a and 14b, we display samples of open-ended responses and multiple-choice questions for the visual perception task. Additionally, Figures 15a and 15b showcase samples of multiple-choice questions and open-ended responses for the visual reasoning task. The focus of the questions differs across tasks. For instance, the visual perception task typically emphasizes image content recognition and understanding, whereas the visual reasoning task places more focus on the ability to draw conclusions from the image. To ensure the accuracy of the evaluation, we select samples that maintain as much independence between tasks as possible, minimizing interference between different tasks. This independence allows each task to reflect the model's performance in a specific task type, thereby providing a more accurate assessment of the model's performance across various tasks and its ability to balance performance between different task types.

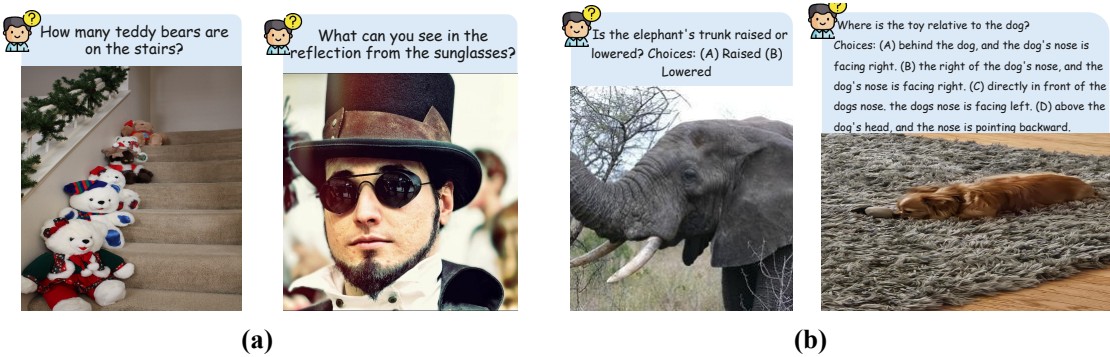

(a)      (b)

Figure 14: Examples of visual perception in *RH-Bench*.

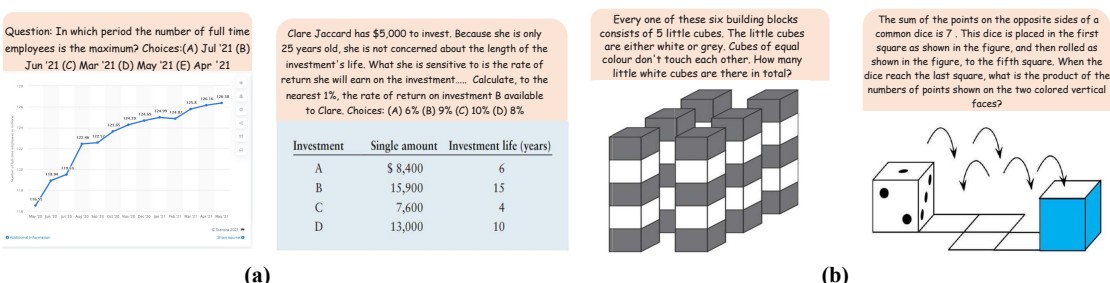

(a)      (b)

Figure 15: Examples of visual reasoning in *RH-Bench*.

# E  Visualization Results

This section presents the responses of reasoning models and their corresponding non-reasoning models across different visual tasks. As shown in Figures 16 and 17, in these error examples, it is evident that reasoning models typically exhibit errors during the reasoning process, leading to inaccurate reasoning results and, consequently, the emergence of hallucinations in the final answers.

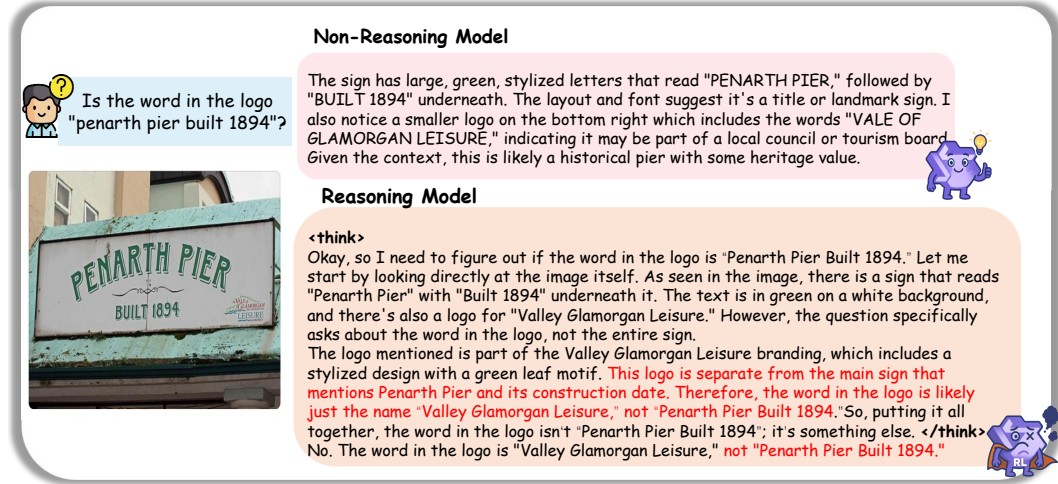

Figure 16: Hallucination example 1 of reasoning models in visual tasks.

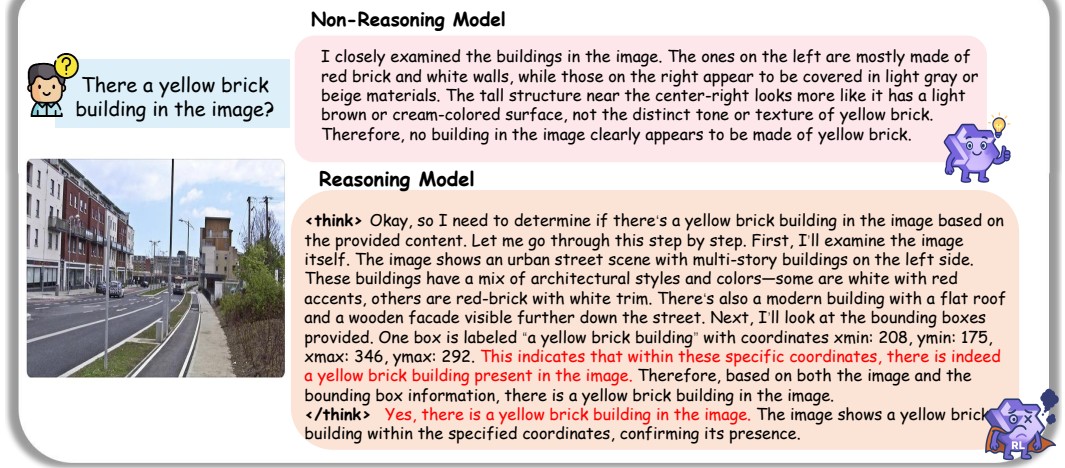

Figure 17: Hallucination example 2 of reasoning models in visual tasks.

# F  *RH-Bench* Evaluation Prompts

---

**RH-Bench Visual Reasoning MCQ & Open-Ended Prompt**

**MCQ:** You are an impartial evaluator assessing the correctness of a model's answer to a multiple-choice question.

Question: {question}
Choices: {choices}
Model's Answer: {model answer}
Correct Answer: {ground truth}

Please evaluate whether the model's answer is correct by considering:
1. Whether the model's answer matches the correct answer exactly (e.g., same option letter).
2. If the model's answer is a value, whether it matches the value of the correct option.
3. Whether the model's reasoning (if provided) supports its answer.

Your response should be a JSON object with the following structure:
{
    "is_correct": <true or false>,
    "reason": "<brief explanation of your evaluation>"
}

**Open-Ended:** You are an impartial evaluator assessing the correctness of a model's answer to a multiple-choice question.

Question: {question}
Model's Answer: {model answer}
Correct Answer: {ground truth}

Please evaluate whether the model's answer is correct by considering:
1. Whether the model's answer matches the correct answer exactly (e.g., same option letter).
2. If the model's answer is a value, whether it matches the value of the correct option.
3. Whether the model's reasoning (if provided) supports its answer.

---

**RH-Bench Visual Perception MCQ Prompt**

Please evaluate whether the model's answer to the multiple-choice question is correct by considering: 1. Whether the model's answer matches the correct answer exactly (same option letter).
2. If the model's answer is a value, whether it matches the value of the correct option.
3. Whether the model's reasoning (if provided) supports its answer.

Question: {}
Options: {}
Correct Answer: {}
Model's Answer: {}
Your response should be a JSON object with the following structure:
{
    "is_correct": <boolean>,
    "reason": "<explanation of your evaluation>",
    "model_answer_extracted": "<the extracted answer from the model's response>"
}

Please act as an impartial and objective judge to evaluate the presence and severity of hallucination in the response provided by a Large Multimodal Model (LMM) to the user question. Hallucination, in this context, refers to a situation where the LMM generates a response that includes information not present or implied in the image or previous conversation. A hallucination could be a false claim about an object, action, emotion, or any other detail not grounded in the image.

Your task is to determine whether hallucination exists and, if present, to categorize its severity based on the extent and impact of the false information. Use the provided image contents, question, standard human-generated answer, and LMM response to make your judgment.

Evaluation Guidelines:
Compare the LMM response with the standard human-generated answer. Identify any claims in the LMM response that are not supported by the image or question. Assess the severity of hallucination based on the nature and extent of the false information. The standard human-generated answer may not describe all objects or details comprehensively, so focus on whether the LMM introduces false claims rather than omitting details. Detailed reasoning or analysis by the LMM that is consistent with the image contents is acceptable and should not be considered hallucination.

Task: Evaluate the LMM response for hallucination based on the provided image contents, question, and standard human-generated answer. Provide a brief explanation of your analysis, identifying any false claims and their severity. Then, assign a rating from the following six levels:

Rating: 5: No hallucination (response is fully consistent with image contents).
Rating: 4: Minimal hallucination (minor, negligible false detail that does not significantly alter the response's accuracy, e.g., a slight misdescription of color or background).
Rating: 3: Mild hallucination (small false claim that adds unsupported but non-critical detail, e.g., mentioning a minor object or attribute not present).
Rating: 2: Moderate hallucination (clear false claim that affects the response's accuracy, e.g., incorrect object count or unsupported environmental detail).
Rating: 1: Severe hallucination (significant false claim that misrepresents key elements, e.g., entirely wrong objects or actions).
Rating: 0: Extreme hallucination (response is largely or entirely detached from the image contents, with multiple or critical false claims).

Question {}
Ground-Truth Answer {}
LMM Response to Evaluate {}
Your response should be a JSON object with the following structure:
{
    "hallucination_score": <integer from 0 to 5>,
    "reason": "<brief explanation of your evaluation>"
}

