# OpenReview forum: "More Thinking, Less Seeing? Assessing Amplified Hallucination in Multimodal Reasoning Models"
_NeurIPS.cc/2025/Conference — NeurIPS 2025 poster_

### Official Review · Reviewer_qZ13 · 2025-06-02

**Clarity:** 4
**Significance:** 4
**Originality:** 4
**Rating:** 5
**Confidence:** 5

**Summary:**

This paper investigates the trade-off between reasoning and visual grounding in multimodal large language models (MLLMs), showing that while extended reasoning chains enhance task performance, they often amplify visual hallucinations due to reduced attention to image inputs and increased reliance on language priors. To quantify this trade-off, the authors introduce RH-AUC, a novel metric that captures the balance between reasoning accuracy and hallucination across varying reasoning lengths. They also release RH-Bench, a diagnostic benchmark comprising 1,000 reasoning and perception tasks to evaluate model performance holistically. Using attention analysis, the study finds that hallucination increases with longer reasoning chains, but the relationship is non-monotonic and task-dependent. Moreover, models trained with reinforcement learning (RL-only) outperform those using supervised fine-tuning plus RL (SFT+RL) in maintaining this balance, and the quality and domain of training data are more influential than data volume in achieving robust visual grounding.

**Questions:**

1- Could the authors elaborate on the specific aspects of the R1-OneVision training paradigm that may limit its ability to balance reasoning and perception, despite its access to extensive visual perception data? For example, are there architectural constraints, insufficient multi-modal supervision signals, or imbalanced task objectives contributing to this outcome?

2- Given that RH-Bench includes diverse tasks beyond mathematics, could the authors clarify whether the observed RH-AUC gains for math-trained models (e.g., MM-Eureka, MM-R1) reflect general improvements in perception-reasoning balance, or are primarily driven by performance on math-like subsets of the benchmark? Would domain-wise breakdowns of RH-AUC help clarify this?

**Ethical Concerns:**

["NO or VERY MINOR ethics concerns only"]

**Final Justification:**

The auhtors addressed my concerns and thus I maintain my positive score.

**Limitations:**

The paper makes a strong and valuable contribution by introducing a benchmark and analysis framework for evaluating perception-reasoning balance in multi-modal models. The overall framing is thoughtful and well-executed. However, some limitations could be more clearly acknowledged. For instance, the choice of non-reasoning baseline is somewhat narrow, which may affect the generalizability of certain claims about grounding behavior. Additionally, a few interpretive points—such as how model training paradigms affect performance or how domain-specific data impacts general reasoning—would benefit from further clarification. These are not critical weaknesses, but addressing them would improve the transparency and depth of the work. Overall, the paper is solid and deserves acceptance.

**Quality:**

4

**Strengths And Weaknesses:**

### Strengths

1- Strong and Actionable Conclusions: The paper draws clear conclusions from its analyses, showing that longer reasoning can lead to more hallucination and that the best reasoning length depends on the task. These findings offer useful guidance for designing and fine-tuning multimodal reasoning models.

2- Strong Benchmark and Metric: The authors introduce RH-Bench and RH-AUC, which together offer a solid and practical way to measure how well models balance reasoning with staying grounded in the image.

3- Clear and Easy to Follow: The paper is well-organized and clearly written, with helpful figures and examples that make even complex ideas, like reasoning length control, easy to understand.

### Weaknesses

1- Narrow Choice of Non-Reasoning Baselines: The paper primarily uses Qwen2.5-VL as the non-reasoning baseline across all comparisons, limiting the diversity of grounding-focused models in the study. This makes it unclear whether the observed hallucination trends generalize across other non-reasoning MLLMs such as LLaVA, or MiniGPT-4, which may have different visual grounding behaviors and could serve as stronger or more varied baselines.

---

> ### Author Rebuttal · Authors · 2025-07-30
>
> We would like to express our heartfelt gratitude to the reviewer for recognizing that **our proposed conclusions are valuable and actionable, our benchmark and metric are solid and practical, and our presentation is well-organized**. We sincerely appreciate the insightful questions raised and have addressed them in detail below.
>
> **(W1) Broader Baseline Coverage.**  Thank you for this suggestion. During our research, we observe most current existing multimodal reasoning models often adopt Qwen2.5-VL as a backbone model and fine-tune on it. Therefore, the set of models we evaluated covers the same model series. To further validate our findings, we conduct a broader evaluation of newer reasoning models across various sizes and their corresponding non-reasoning counterparts. As shown in Table 1, the results align with our findings, further corroborating the universality of the hallucination trends and visual grounding behaviors proposed in this paper.
>
> **Table 1: Performance of recently released multimodal reasoning models on perception benchmarks and RH-Bench.**
> | Method             | Train Setting | Training Data | Training Data | MMVP     | MMEval-Pro | VMCBench | MMHALU | MMHALU | RH-Bench |
> |--------------------|---------------|---------------|---------------|----------|------------|----------|--------|--------|--------|
> |                    |               | Perc.         | Reas.         | Acc ↑    | Acc↑        | Acc↑      | Score↑ | Hallu↓ | RH-AUC↑  |
> | **Qwen2-VL-7B**        | Base          | -             | -             | 45.7     | 75.0       | 82.7     | 3.48   | 0.36   | -      |
> | R1-VL [1]          | SFT+RL        | 21k           | 40k           | 43.8     | 73.7       | 81.5     | 3.40   | 0.39   | 0.47   |
> | **InternVL2.5-8B**     | Base          | -             | -             | 45.0     | 74.6       | 81.5     | 3.52   | 0.35   | -      |
> | MM-Eureka [2]      | RL            | 19k           | 38k           | 44.2     | 72.5       | 80.0     | 3.48   | 0.38   | 0.53   |
> | **InternVL2.5-38B**    | Base          | -             | -             | 49.5     | 78.0       | 83.3     | 3.57   | 0.33   | -      |
> | MM-Eureka [2]      | RL            | -             | 9k            | 47.6     | 77.2       | 81.6     | 3.53   | 0.35   | 0.64   |
> | **Qwen2.5-VL-32B**     | Base          | -             | -             | 50.4     | 78.5       | 84.0     | 3.63   | 0.32   | -      |
> | MM-Eureka-Qwen [3] | RL            | -             | 17k           | 48.5     | 77.9       | 82.8     | 3.60   | 0.34   | 0.63   |
> | Vision-R1 [4]      | SFT+RL        | 60k           | 10k           | 47.8     | 77.0       | 81.9     | 3.56   | 0.36   | 0.60   |
> | **Qwen-2.5-VL-72B**    | Base          | -             | -             | 51.0     | 79.3       | 86.1     | 3.70   | 0.30   | -      |
> | VL-Rethinker [5]   | RL            | 6k            | 14k           | 49.7     | 77.8       | 84.9     | 3.67   | 0.33   | 0.68   |
> | Vision-R1 [4]      | SFT+RL        | 60k           | 10k           | 50.8     | 78.0       | 84.8     | 3.63   | 0.37   | 0.65   |
>
> **(Q1) Factors Influencing The Reasoning-Perception Balance of R1-OneVision.**  Thank you for the insightful question. The performance limitations of the R1-OneVision model, built upon the Qwen2.5-VL architecture, are not attributable to inherent architectural constraints of the base model. This is evidenced by the fact that numerous other superior reasoning models also utilize the same baseline. The internal attention mechanism analysis of the R1-OneVision model, as depicted in Fig. 5 of the manuscript, reveals a significant dilution effect on visual information during the reasoning process. This suggests a potential imbalance in the task objectives of R1-OneVision, with the optimization focus skewed towards enhancing reasoning capabilities. Furthermore, the adoption of the SFT+RL training paradigm without integrating additional optimization strategies likely contributes to an imbalance between reasoning and perception abilities.
>
> **(Q2) Domain-wise Analysis of RH-AUC Performance.**  Thank you for highlighting this. To investigate the RH-AUC improvement of math-trained models (e.g., MM-Eureka and MM-R1) on RH-Bench, we categorize the benchmark into math and non-math tasks and separately evaluate the accuracy and RH-AUC scores of each model. As shown in Table 2, the results demonstrate that MM-Eureka and MM-R1 indeed achieve higher RH-AUC scores in pure mathematics tasks, clearly reflecting their superior perception-reasoning balance capability within the mathematical domain. Notably, even on non-math tasks, the RH-AUC of MM-Eureka and MM-R1 surpasses that of some models with extensive training data, such as R1-OneVision. This suggests domain-specific training holds the potential to generalize to other domains, improving perception-reasoning balance, which aligns with our paper's findings.
>
> **Table2: Comparison of RH-AUC across math and non-math domains for different models.**
>
> | Method              | Math | Non-Math |
> |---------------------|------|----------|
> | MM-Eureka-7B        | 0.58 | 0.54     |
> | MM-R1-7B            | 0.59 | 0.55     |
> | R1-Onevision-7B     | 0.45 | 0.49     |
> | ThinkLite-VL-7B     | 0.51 | 0.53     |
>
> **(L1) Further Clarification of Interpretive Points.**  Thank you for your valuable feedback. Regarding the impact of training paradigms and data domains, we present the performance and RH-AUC of various advanced models in Table 1. These results clearly indicate that recent reasoning models no longer rely solely on larger datasets but rather prioritize carefully curated multi-domain data, which aligns with our analysis. Furthermore, most RL-only models exhibit superior performance. For these two training paradigms (RL vs. SFT+RL), our analysis in Tables 3 and 4 below conducts fine-grained statistics on response length and reasoning-related vocabulary. The results reveal a statistically significant tendency of RL models to generate more concise responses while avoiding excessive deliberation, which is consistent with the results in Figs. 8b–c of the manuscript. Moreover, Table 5 presents a more comprehensive comparison of the RL and SFT+RL training paradigms.
>
> **Table3: Average response token length.**
> | Benchmark  | Baseline (k) | SFT + RL (k) | RL (k) |
> |------------|--------------|--------------|--------|
> | MathVision | 0.70         | 4.8          | 0.8    |
> | MathVista  | 0.35         | 3.2          | 0.38   |
> | MMVP       | 0.20         | 1.80         | 0.23   |
> | MMHalu     | 0.13         | 1.72         | 0.16   |
>
> **Table4: Frequency of reasoning related words evaluated on MathVision.**
> | Word              | Baseline | SFT + RL | RL   |
> |-------------------|----------|----------|------|
> | `"wait"`          | 850      | 3205     | 2353 |
> | `"check"`         | 783      | 2040     | 1780 |
> | `"alternative"`   | 566      | 1204     | 800  |
> | `"however"`       | 620      | 1532     | 980  |
>
> **Table5: Performance comparison of two training paradigm models on multimodal benchmarks.**
> | Method                | Paradigms | MMVP        | MMEval-Pro  | VMCBench    | MMHALU | MMHALU | RH-Bench | RH-Bench  |
> |-----------------------|-----------|-------------|-------------|-------------|--------------|--------------|----------------|----------------|
> |                       |           | Acc ↑       | Acc ↑       | Acc ↑       | Score↑           |  Hallu↓            |    Perc.↑             |    Reas.↑            |
> | SophiaVL-7B [6]       | SFT+RL    | 45.0        | 73.0        | 80.2        | 3.25         | 0.45         | 58.7           | 52.3           |
> | ReVisual-R1-7B [7]    | SFT+RL    | 46.7        | 72.5        | 80.0        | 3.30         | 0.44         | 60.2           | 53.5           |
> | VLAA-Thinker-7B [8]   | RL        | 47.4        | 75.7        | 81.3        | 3.46         | 0.41         | 64.5           | 56.4           |
> | MM-Eureka-Qwen-7B [3] | RL        | 46.7        | 74.8        | 82.0        | 3.43         | 0.43         | 62.4           | 54.0           |
>
> [1] *R1-VL: Learning to Reason with Multimodal Large Language Models via Step-wise Group Relative Policy Optimization*
>
> [2] *MM-Eureka: Exploring visual aha moment with rule-based large-scale reinforcement learning*
>
> [3] *MM-Eureka: Exploring the frontiers of multimodal reasoning with rule-based reinforcement learning*
>
> [4] *Vision-R1: Incentivizing Reasoning Capability in Multimodal Large Language Models*
>
> [5] *VL-Rethinker: Incentivizing Self-Reflection of Vision-Language Models with Reinforcement Learning*
>
> [6] *SophiaVL-R1: Reinforcing MLLMs Reasoning with Thinking Reward*
>
> [7] *Advancing Multimodal Reasoning: From Optimized Cold Start to Staged Reinforcement Learning*
>
> [8] *SFT or RL? An Early Investigation into Training R1-Like Reasoning Large Vision-Language Models*

---

> ### Comment · Reviewer_qZ13 · 2025-08-03
>
> Thank you for the extensive response. I am satisfied with the answers and will maintain my positive score.

---

> > ### Author Response · Authors · 2025-08-03
> >
> > Thank you for taking the time to review our responses and your recognition of our paper. Your feedback and support have been invaluable to our work.

---

### Official Review · Reviewer_cecd · 2025-07-01

**Clarity:** 4
**Significance:** 2
**Originality:** 3
**Rating:** 4
**Confidence:** 4

**Summary:**

This paper studies the hallucination observed in recent reasoning VLMs. The authors begin with a preliminary analysis showing that compared to the original VLMs, reasoning VLMs with reasoning chains tend to produce more perception-related hallucinations. Through attention-based analysis, they identify that reasoning models tend to reduce visual attention over time, which leads to the rise in hallucination. Further investigation reveals a non-monotonic relationship between reasoning length and  reasoning-perception performance.

To quantify the trade-off between reasoning and hallucination in VLMs, the authors propose: (1) RH-AUC, a metric that measures the perception accuracy as reasoning length varies; and (2) RH-Bench, a diagnostic benchmark designed to evaluate both reasoning and hallucination across diverse multimodal tasks. Experimental results show that larger models generally achieve a better balance between reasoning and hallucination, and that the types and domains of training data significantly influence the performance.

**Questions:**

## Questions

In addition to the concerns raised in the weaknesses section:
- Have you considered incorporating larger VLMs (e.g., GPT-4) into your experiments to assess how their accuracy varies with different reasoning chain lengths? This could provide a broader perspective on the reasoning–hallucination trade-off.
- In the results presented in Section 5, which length-control methods (e.g., Token Budget Forcing, Test-Time Scaling, Latent State Steering) were used to generate reasoning chains of different lengths? Does the choice of length-control method impact model performance on RH-AUC? Is it possible that a model achieves a low RH-AUC due to the control method rather than the model’s actual capabilities?

## Writing
- Line 67: missing space after the comma.


*Note: I am currently leaning toward a borderline rejection. However, I am open to increasing my score if the concerns outlined above are adequately addressed during the rebuttal phase.*

**Ethical Concerns:**

["NO or VERY MINOR ethics concerns only"]

**Final Justification:**

The author responses have addressed most of my concerns. Overall, I consider the work interesting and relevant to a timely problem. One remaining question lies with W1: although the rebuttal includes comparisons across various VLMs to assess the influence of SFT and RL training, these models may differ in terms of training data, implementation details, and underlying paradigms. As such, the conclusions drawn may not be entirely robust.

In general, I appreciate the authors' efforts and will raise my score to a 4.0 as borderline accept.

**Limitations:**

yes

**Quality:**

3

**Strengths And Weaknesses:**

## Strengthes
- This paper addresses a timely and important problem concerning hallucination in recent reasoning VLMs. The observations and conclusions presented may offer insights for future research in the field. The overall motivation and logical flow of the paper are clear and well-structured.
- The introduction of RH-AUC and RH-Bench allows for a nuanced quantification of the trade-off between reasoning and visual grounding, enabling a more comprehensive evaluation of recent VLMs on this issue.
- The paper conducts comparisons across multiple recent models with varying sizes (e.g., 3B and 7B) and training strategies (e.g., SFT and RL), offering insights that could benefit the development of future models.

## Weaknesses

- Some of the conclusions and analyses regarding data domains, training paradigms, and model performance are primarily observational, based on off-the-shelf models, without the support of rigorously controlled experiments (e.g., training models with varied data or training strategies). As a result, certain claims—such as “RL only outperforms SFT+RL” or “data domain matters more than data volume”—may be less convincing.
- While RH-AUC enables detailed measurement of performance across reasoning chain lengths, it comes with high computational costs, potentially limiting its practical usability. Moreover, enforcing strict control over reasoning length can be rigid, especially given that current reasoning VLMs are not trained with explicitly controllable reasoning depth. For example, a model might naturally prefer shorter reasoning chains to achieve optimal answers. A potentially more realistic evaluation would allow models to generate multiple diverse reasoning chains per question, assessing performance without prescribing specific lengths.
- A related concern is that more complex questions often elicit longer reasoning chains. However, RH-Bench does not appear to incorporate this factor into its design, which may limit its effectiveness in accurately evaluating model performance across varying question difficulties.
- The current analysis is primarily conducted on models from the Qwen2.5-VL (as noted in the limitations), which may constrain the generalizability of the findings to broader classes of VLMs.

---

> ### Author Rebuttal · Authors · 2025-07-31
>
> We sincerely thank the reviewer for recognizing **our observations as timely, insightful, and important, our comprehensive quantitative evaluation, and our systematic analysis of the reasoning–hallucination trade-off**. We sincerely hope that our responses below address your concerns.
>
> **(W1) Empirical Analysis and Generalizability Validation.**  Thank you for pointing this out. We agree that our current analyses are based on off-the-shelf models, and while they offer valuable insights, they should be interpreted as observational rather than causal,  this is indeed what we have done in the paper where we show our findings suggest trends and raise hypotheses, such as “RL only outperforms SFT+RL”, we will make this even clearer in the revised version and warrant future controlled studies. To further demonstrate the generalizability of our findings, we conduct a comprehensive performance evaluation of recently released reasoning models. As shown in Table 1 of **qZ13 W1**, the results not only reveal a consistent trend of amplified hallucinations but also further support the finding that training data volume is not the sole determinant of model performance. As shown in Table 1 below, the performance comparisons across different training paradigms, along with the statistical analyses of reasoning length and reasoning-related terms presented in Tables 2–3, are consistent with the findings reported in our manuscript.
>
> **Table1: Performance comparison of two training paradigm models on multimodal benchmarks.**
> | Method                | Paradigms | MMVP        | MMEval-Pro  | VMCBench    | MMHALU | MMHALU | RH-Bench | RH-Bench  |
> |-----------------------|-----------|-------------|-------------|-------------|--------------|--------------|----------------|----------------|
> |                       |           | Acc ↑       | Acc ↑       | Acc ↑       | Score↑           |  Hallu↓            |    Perc.↑             |    Reas.↑            |
> | SophiaVL-7B [1]       | SFT+RL    | 45.0        | 73.0        | 80.2        | 3.25         | 0.45         | 58.7           | 52.3           |
> | ReVisual-R1-7B [2]    | SFT+RL    | 46.7        | 72.5        | 80.0        | 3.30         | 0.44         | 60.2           | 53.5           |
> | VLAA-Thinker-7B [3]   | RL        | 47.4        | 75.7        | 81.3        | 3.46         | 0.41         | 64.5           | 56.4           |
> | MM-Eureka-Qwen-7B [4] | RL        | 46.7        | 74.8        | 82.0        | 3.43         | 0.43         | 62.4           | 54.0           |
>
> **Table2: Average response token length.**
> | Benchmark  | Baseline (k) | SFT + RL (k) | RL (k) |
> |------------|--------------|--------------|--------|
> | MathVision | 0.70         | 4.85          | 0.81    |
> | MathVista  | 0.35         | 3.23         | 0.38   |
> | MMVP       | 0.20         | 1.80         | 0.23   |
> | MMHalu     | 0.13         | 1.72         | 0.16   |
>
> **Table3: Frequency of reasoning related words evaluated on MathVision.**
> | Word              | Baseline | SFT + RL | RL   |
> |-------------------|----------|----------|------|
> | `"wait"`          | 850      | 3205     | 2353 |
> | `"check"`         | 783      | 2040     | 1780 |
> | `"alternative"`   | 566      | 1204     | 800  |
> | `"however"`       | 620      | 1532     | 980  |
>
> **(W2) Explanation of RH-AUC Computation.**   Thank you for your insightful suggestion. We strongly agree with the view that models should be able to generate diverse reasoning chains per question, and our **Latent State Steering** method (Section 4.1) is specifically designed to achieve this capability. This method enables flexible control over diverse reasoning chain generation through a single tunable parameter $\alpha$, dynamically adjusting reasoning length without fixed constraints. Compared to direct generation of diverse reasoning chains, which often yields limited length variation and narrow scope, our steering method achieves significantly greater diversity through $\alpha$ adjustment while preserving the model's original performance (see detailed analysis in Tables 4–5). Appendix Figs. 9–11 show the performance under our steering method. Notably, this method requires only a single hyperparameter adjustment without introducing additional computational overhead.
>
> **(W3) Relationship Between Reasoning Length and Problem Complexity.**  We thank the reviewer for raising this important point. It is true that, in practice, more complex questions may naturally elicit longer reasoning chains. However, RH-Bench is designed to decouple reasoning length from question difficulty in order to evaluate the model’s robustness to reasoning depth under controlled settings. To minimize confounding effects, we fix the question while varying only the reasoning chain length via intervention, allowing us to probe how well the model performs as reasoning length increases even for the same question. This design choice helps us isolate the impact of reasoning length itself, rather than entangling it with question difficulty. We acknowledge this limitation and have clarified this design rationale in Section 5, while also noting that evaluating models under naturally varying question difficulty remains an important direction for complementary future benchmarks.
>
> **(W4) Diversity of Base Models.**  Thank you for your valuable feedback. We respectfully note that most current existing multimodal reasoning models often adopt Qwen2.5-VL as a backbone model and fine-tune on it. Therefore, the set of models we evaluated covers the same model series. To further demonstrate generalizability, we conduct additional experiments on recently released reasoning models and baselines: Qwen2-VL (7B), InternVL2.5 (8B, 38B), and Qwen2.5-VL (32B, 72B). Detailed results are shown in Table 1 of **qZ13 W1**.
>
> **(Q1) Reasoning–Hallucination Trade-off in Larger VLMs.**  Closed-source models like GPT-4 and O3 struggle to precisely control reasoning chain length, relying mainly on limited prompt engineering. In contrast, our systematic evaluation of advanced open-source reasoning models (32B to 72B) in Table 1 of **qZ13 W1** demonstrates that larger models typically achieve higher RH-AUC scores, indicating more robust performance.
>
> **(Q2) Length-Control Method Impact on RH-AUC Performance.**  Thank you very much for your thoughtful suggestions. The RH-AUC evaluation employs the Latent State Steering method, with its implementation details shown in Appendix D. Furthermore, we conduct comparative experiments on the RH-Bench benchmark using three length control methods to assess their impact on model performance. As shown in Tables 4–5 below, by analyzing the accuracy of perception and reasoning tasks and the RH-AUC metrics of different models, the results indicate that the performance differences among the methods on these tasks fluctuate around ±0.5%. Moreover, the RH-AUC metrics demonstrate high stability across all three methods. These results indicate that the reasoning length control methods only affect the output length without compromising the model's performance.
>
> **Table4: Accuracy (%) comparison on RH-Bench across three reasoning length control methods: Token Budget Forcing, Test Time Scaling, and Latent State Steering (LSS).**
>
> | Think Length Range |                |       Perception           |                   |                  |       Reason             |              |
> |--------------|--------------------------|--------------------|-------------------|--------------------------|------------------|--------------|
> |                   | Scaling                  | Budget             | LSS               | Scaling                  | Budget           | LSS          |
> | 50–150       | 54.8                     | 54.5               | 55.0              | 45.4                     | 45.2             | 45.5         |
> | 151–250      | 56.0                     | 56.2               | 56.5              | 46.3                     | 46.0             | 46.4         |
> | 251–350      | 56.5                     | 56.0               | 56.8              | 47.8                     | 47.5             | 48.0         |
> | 351–450      | 57.6                     | 57.3               | 57.8              | 48.6                     | 48.4             | 48.9         |
> | 451–550      | 55.8                     | 55.6               | 56.1              | 48.1                     | 48.3             | 48.6         |
>
> **Table5: RH-AUC scores across different models under three reasoning length control methods.**
>
> | Model                 | Scaling | Budget | LSS   |
> |-----------------------|---------|--------|-------|
> | LLM-R1-3B             | 0.462   | 0.464  | 0.465 |
> | Curr-ReFT-3B          | 0.473   | 0.470  | 0.473 |
> | R1-OneVision-7B       | 0.465   | 0.465  | 0.467 |
> | ThinkLite-VL-7B       | 0.525   | 0.523  | 0.526 |
> | MM-Eureka-Qwen-7B     | 0.550   | 0.550  | 0.552 |
> | MM-Eureka-Qwen-32B    | 0.609   | 0.607  | 0.610 |
> | VL-Rethinker-72B [5]      | 0.635   | 0.632  | 0.635 |
>
> **(Q3) Correct Formatting.**  Thank you for highlighting this. We have carefully reviewed the text and made the necessary corrections in the revised manuscript.
>
>
> [1] *SophiaVL-R1: Reinforcing MLLMs Reasoning with Thinking Reward*
>
> [2] *Advancing Multimodal Reasoning: From Optimized Cold Start to Staged Reinforcement Learning*
>
> [3] *SFT or RL? An Early Investigation into Training R1-Like Reasoning Large Vision-Language Models*
>
> [4] *MM-Eureka: Exploring the Frontiers of Multimodal Reasoning with Rule-Based Reinforcement Learning*
>
> [5] *VL-Rethinker: Incentivizing Self-Reflection of Vision-Language Models with Reinforcement Learning*

---

> > ### Comment · Reviewer_cecd · 2025-08-04
> >
> > Thank you for your detailed response, which has addressed most of my previous concerns and questions. Overall, I consider the work interesting and relevant to a timely problem. One remaining question lies with W1: although the rebuttal includes comparisons across various VLMs to assess the influence of SFT and RL training, these models may differ in terms of training data, implementation details, and underlying paradigms. As such, the conclusions drawn may not be entirely robust.
> >
> > In general, I appreciate the authors' efforts and will raise my score to a 4.0 as borderline accept.

---

> > > ### Author Response · Authors · 2025-08-05
> > >
> > > Thank you for taking the time to review our rebuttal and raising the score. Your comments have significantly helped improve our work. We will include the rebuttal in the final version and consolidate our response accordingly.

---

### Official Review · Reviewer_u6Et · 2025-07-02

**Clarity:** 3
**Significance:** 3
**Originality:** 2
**Rating:** 4
**Confidence:** 3

**Summary:**

This paper focuses on the phenomenon that multimodal reasoning models tend to exhibit more hallucinations in perception tasks compared to non-reasoning models. Through empirical studies, they attribute this to reasoning models relying more on language priors and paying less attention to visual input, which causes hallucinations. Furthermore, the paper finds that the optimal reasoning intervals for minimizing hallucinations differs from that for maximizing reasoning performance, and existing metrics fail to capture the balance between hallucination and reasoning ability. To better evaluate this trade-off, the authors propose the RH-AUC metric and the RH-Bench dataset, providing insights based on results from multiple reasoning models.

**Questions:**

1. I recommend a deeper discussion on the soundness of the Latent State Steering operation.
2. It would be beneficial to include additional empirical analysis of the training paradigms to provide more valuable insights into the training of multimodal reasoning models.
3. A clearer explanation of the RH-AUC metric, along with a comprehensive analysis of its advantages, would greatly enhance the paper.
4. The results presented in Figure 6a of the appendix appear somewhat counterintuitive: the reasoning model exhibits fewer perception errors but more reasoning errors compared to the non-reasoning model, which seems to contradict the observations highlighted in the main text. Could a plausible explanation for this discrepancy be provided?

**Ethical Concerns:**

["NO or VERY MINOR ethics concerns only"]

**Final Justification:**

While I still regard the technical and conceptual contributions as somewhat incremental, and believe that the conclusions regarding the training paradigm would benefit from further substantiation (I emphasize this point as I consider it a key aspect of the paper’s potential value), I appreciate that the additional experiments—particularly those presented in the responses to other reviewers—have further strengthened the overall quality and contribution of the work. Accordingly, I am raising my score to 4.

**Limitations:**

yes

**Paper Formatting Concerns:**

There are no formatting issues.

**Quality:**

3

**Strengths And Weaknesses:**

### Strengths
1. The paper focuses on an important issue that multimodal reasoning models are more prone to hallucination. Through a systematic analysis across a range of advanced models, it offers valuable insights, including that models trained with SFT followed by RL are most susceptible to hallucination.
2. The evaluation of multimodal reasoning models should account for both reasoning ability and hallucination severity. This work introduces a metric designed to reflect both factors.
3. The paper is well-written and clearly structured, making it easy to follow.

### Weaknesses
1. **Limited Novelty in Attribution Analysis.** Prior work, such as [1, 2], has already highlighted the over-reliance on language priors as a key contributor to hallucinations. The observations in this work—such as weak visual attention and declining visual focus over longer reasoning chains—seem to be natural extensions of these findings rather than fundamentally new insights.
2. **Lack of a Technical Contribution.** While the paper raises an important and timely question, it does not propose a concrete technical method or solution, which constrains its contribution.
3. **Limited Empirical Support for Training Paradigms Claims.** The paper’s claim that SFT leads to rigid imitation reasoning and redundant steps would benefit from more direct empirical validation.
4. **Concerns About the Methodological Soundness of Latent State Steering.** Since the outputs are generated from an intervened latent state rather than the model’s natural reasoning process, it remains unclear how well they reflect the model’s true capabilities. This raises questions about the validity of key analyses (Figure 6 and 7) and the RH-AUC metric that rely on this technique.
5. **Interpretability and Effectiveness of the RH-AUC Metric.** The RH-AUC metric correlates strongly with both perception and reasoning performance. This suggests it may reflect overall performance rather than specifically measuring the balance between reasoning and perception. More analysis is needed to demonstrate its advantage over using perception and reasoning accuracy directly.

[1] Mitigating Object Hallucinations in Large Vision-Language Models through Visual Contrastive Decoding
[2] Paying More Attention to Image: A Training-Free Method for Alleviating Hallucination in LVLMs

---

> ### Author Rebuttal · Authors · 2025-07-30
>
> We thank the reviewer for the valuable feedback, and for recognizing **the importance of our proposed problem, the insightful findings provided by our systematic analysis, and the clarity and readability of our paper’s presentation**. We sincerely hope our response below can address your concerns.
>
> **(W1) Discussion with Relevant Works.**  Thank you for highlighting these relevant works. We will include the suggested discussion in the revised manuscript. While we indeed follow the attention analysis approach from prior work, the focus of our paper is different. We are not attempting to explain the general source of hallucination in language models; rather, we aim to explain how hallucination in reasoning language models is amplified. Specifically, our attention analysis demonstrates that reasoning models produce longer reasoning chains, which consequently leads to reduced attention toward visual inputs.
>
> **(W2) Concrete Technical Contribution.**  Thank you for bringing up this point. We indeed proposed several techniques to balance the hallucination and reasoning ability of multimodal reasoning language models. Specifically, the **Latent State Steering** method (mentioned in Sec. 4.1 of the manuscript) is designed to mitigate hallucination while maintaining reasoning ability (see detailed analysis in table3 below) by dynamically adjusting latent representations to control the reasoning chain length. As demonstrated in Fig. 9–11 in the Appendix, our approach effectively reduces hallucinations without sacrificing the original reasoning performance of the model.
>
> **(W3 & Q2) Empirical Support for Training Paradigms Claims.**  Thank you for your valuable suggestion. We agree that some more direct validation would make the argument stronger, and we demonstrated some preliminary analysis in the paper: in Fig. 8b, we demonstrate that SFT+RL may increase reasoning length. Moreover, we illustrate differences in reasoning patterns between these paradigms through a case study presented in Fig. 8c. To further investigate the difference in reasoning behavior between SFT and SFT+RL models, we compare the average token length and the frequency of reasoning-related vocabulary in responses from two model paradigms. As the results in the Table 1 and 2 indicate, SFT+RL models generate significantly more redundant responses than RL models, which aligns with our findings.
>
> **Table1: Average response token length.**
> | Benchmark  | Baseline (k) | SFT + RL (k) | RL (k) |
> |------------|--------------|--------------|--------|
> | MathVision | 0.70         | 4.85          | 0.81    |
> | MathVista  | 0.35         | 3.23       | 0.38   |
> | MMVP       | 0.20         | 1.80         | 0.23   |
> | MMHalu     | 0.13         | 1.72         | 0.16   |
>
> **Table2: Frequency of reasoning related words evaluated on MathVision.**
> | Word              | Baseline | SFT + RL | RL   |
> |-------------------|----------|----------|------|
> | `"wait"`          | 850      | 3205     | 2353 |
> | `"check"`         | 783      | 2040     | 1780 |
> | `"alternative"`   | 566      | 1204     | 800  |
> | `"however"`       | 620      | 1532     | 980  |
>
> **(W4 & Q1) Validation and Analysis of Latent State Steering.**  Thank you for this suggestion. We would like to clarify that our proposed steering method does not compromise the model's natural reasoning capabilities. Based on the results presented in Table 3–4 below, we evaluate the performance of three length control methods: Token Budget Forcing, Test Time Scaling, and Latent State Steering on RH-Bench. The results show that when controlling for the same length range, the differences in the impact of these methods on model performance in perception and reasoning tasks are minimal. This suggests that the observed performance variations are mostly due to the reasoning length, rather than the specific control method. Furthermore, our steering method uses the hyperparameter $\alpha$ to control output lengths, where setting $\alpha=0$ allows the model to reason without intervention.
>
> **Table3: Accuracy (%) comparison on RH-Bench across three reasoning length control methods: Token Budget Forcing, Test Time Scaling, and Latent State Steering (LSS).**
>
> | Think Length |                |       Perception           |                   |                  |       Reason             |              |
> |--------------|--------------------------|--------------------|-------------------|--------------------------|------------------|--------------|
> |                   | Scaling                  | Budget             | LSS               | Scaling                  | Budget           | LSS          |
> | 50–150       | 54.8                     | 54.5               | 55.0              | 45.4                     | 45.2             | 45.5         |
> | 151–250      | 56.0                     | 56.2               | 56.5              | 46.3                     | 46.0             | 46.4         |
> | 251–350      | 56.5                     | 56.0               | 56.8              | 47.8                     | 47.5             | 48.0         |
> | 351–450      | 57.6                     | 57.3               | 57.8              | 48.6                     | 48.4             | 48.9         |
> | 451–550      | 55.8                     | 55.6               | 56.1              | 48.1                     | 48.3             | 48.6         |
>
> **Table4: RH-AUC scores across different models under three reasoning length control methods.**
>
> | Model                 | Scaling | Budget | LSS   |
> |-----------------------|---------|--------|-------|
> | LLM-R1-3B             | 0.462   | 0.464  | 0.465 |
> | Curr-ReFT-3B          | 0.473   | 0.470  | 0.473 |
> | R1-OneVision-7B       | 0.465   | 0.465  | 0.467 |
> | ThinkLite-VL-7B       | 0.525   | 0.523  | 0.526 |
> | MM-Eureka-Qwen-7B     | 0.550   | 0.550  | 0.552 |
> | MM-Eureka-Qwen-32B [1]   | 0.609   | 0.607  | 0.610 |
> | VL-Rethinker-72B [2]      | 0.635   | 0.632  | 0.635 |
>
>
> **(W5 & Q3) Interpretability Analysis and Evaluation Advantages of the RH-AUC.**  Thank you for your insightful comments. We would like to clarify that RH-AUC is a metric designed to assess the interaction between perception and reasoning across varying reasoning lengths, reflecting an overall balance. This metric is obtained by plotting perception accuracy (y-axis) against reasoning accuracy (x-axis) at various reasoning lengths. As reasoning length increases, perception accuracy typically declines due to diminished visual attention (as shown in Figures 5–6 of the manuscript). Therefore, the area under this curve captures the overall balance between perception and reasoning.
>
> The advantage of RH-AUC over traditional metrics lies in its ability to assess the interaction between perception and reasoning, rather than just measuring their individual accuracy. A simple average of perception and reasoning accuracy inevitably masks their critical interaction under various reasoning lengths. For example, a model with 0.8 perception accuracy but 0.2 reasoning accuracy, and another model with both scores at 0.5, would have the same average (0.5). However, the second model shows better balance. Traditional metrics fail to reflect this, while RH-AUC effectively evaluates the balance between perception and reasoning by testing performance across different reasoning levels.
>
> **(Q4) Further Analysis of Error Proportions in Reasoning Models.**  Thank you for bringing up this point. The distinction between reasoning and perception errors in Appendix Figs. 6a is as follows: perception errors occur at the initial stage of the response, showing wrong identification of visual content (e.g., misidentifying three dogs as two in an image). In contrast, reasoning errors arise during the reasoning after the model has correctly perceived the content (e.g., accurately recognizing the spatial relationship between a red car and a tree but drawing an incorrect answer about whether the car is parked under the tree). The increase in the proportion of reasoning errors further supports our finding that the decline of attention with longer reasoning chains is the key factor exacerbating hallucination, rather than issues in early-stage visual information processing.
>
> [1] *MM-Eureka: Exploring the frontiers of multimodal reasoning with rule-based reinforcement learning*
>
> [2] *VL-Rethinker: Incentivizing Self-Reflection of Vision-Language Models with Reinforcement Learning*

---

> > ### Author Response · Authors · 2025-08-06
> >
> > Dear Reviewer u6Et,
> >
> > We greatly appreciate your feedback, which has been invaluable in improving our work. We have carefully addressed your comments about the review of relevant works, concrete technical contributions, empirical support for training paradigms, validation and analysis of latent state steering, interpretability and evaluation advantages of RH-AUC, and extended analysis of error proportions in reasoning models.
> >
> > As the discussion period is ending in two days, we kindly invite you to review our response. We hope our response addresses adequately your concerns. Thanks!
> >
> > Best,
> >
> > The Authors

---

> ### Comment · Reviewer_u6Et · 2025-08-06
>
> Thank you for the authors' detailed and thoughtful response. I have carefully read the rebuttal and appreciate the clarifications provided. The responses to W4 and W5 are helpful and seem to address the concerns raised.
>
> Regarding W1, while the authors’ explanation clarifies the contribution, I still feel that the novelty is somewhat incremental. For W2, I would note that latent space steering has been explored in prior works within the LVLM domain (e.g., [1]), and may not be entirely novel. As for W3, I share reviewer cecd’s concern that the conclusions drawn about the training paradigm may not be entirely robust.
>
> [1] Reducing Hallucinations in Vision-Language Models via  Latent Space Steering

---

> ### Author Response · Authors · 2025-08-06
>
> We sincerely thank the reviewer for the thoughtful feedback and for carefully engaging with our rebuttal and would like to take this opportunity to clarify the core contributions and positioning of our work. Our work is the first, to our knowledge, to uncover and systematically analyze the phenomenon of **hallucination amplification in** multimodal **reasoning** models. This issue, where increasing reasoning length exacerbates hallucinations, has not been previously studied in the literature, and we believe it addresses an **urgent and underexplored challenge** in the development of reliable vision-language systems. The **significance and timeliness** of this problem have been positively **recognized by reviewers kC9o, cecd, and qZ13**.
>
> To study this issue rigorously, we introduce a **comprehensive evaluation framework**, including novel metrics such as **RH-AUC** and the **RH-Bench** diagnostic benchmark. These tools enable precise measurement of how reasoning interacts with visual grounding, providing actionable insights into the reasoning-perception tradeoff. **The utility and novelty of this framework have been acknowledged by reviewers qZ13 and cecd**. We are also the first to propose multiple techniques for controlling the **reasoning length** in multimodal reasoning models to investigate the **impact of varying reasoning lengths on the balance of reasoning models**.
>
> **(W2)** We sincerely thank the reviewer for mentioning the work in [1]. We'd like to clarify that our method has a different focus from that of [1]. Our steering method focuses on dynamically adjusting reasoning length to investigate its effect on the reasoning-perception balance. Through this method, we analyze how changes in the length of the reasoning chain impact a model's performance stability on both perception and reasoning tasks. In contrast, the method in [1] primarily addresses the single issue of mitigating hallucinations in multimodal models. Furthermore, we have already cited [1] in our paper and will include a more detailed discussion in our revised version.
>
> **(W3)** We understand the reviewer's concern. We would like to clarify that our paper does not make definitive conclusions about training paradigms. Instead, based on systematic observations of the performance of existing reasoning models, we present analytical findings. In the paper, we used terms such as “may” and “not always” to maintain caution. While these findings are not conclusive, they provide a solid foundation and clear direction for future research.
>
> [1] *Reducing Hallucinations in Vision-Language Models via Latent Space Steering*

---

> > ### Comment · Reviewer_u6Et · 2025-08-09
> >
> > Thank you for the authors’ detailed and thoughtful response. While I still regard the technical and conceptual contributions as somewhat incremental, and believe that the conclusions regarding the training paradigm would benefit from further substantiation (I highlight this point because I consider it an important aspect of the paper’s potential value), I appreciate that the additional experiments—particularly those presented in the responses to other reviewers—have further strengthened the overall quality and contribution of the work. Accordingly, I am raising my score to 4.

---

> > > ### Author Response · Authors · 2025-08-09
> > >
> > > Thank you for taking the time to review our rebuttal and for raising the score. Your comments, along with the recognition of the strengthened contributions through our additional experiments, have been invaluable in refining our work. We will incorporate the rebuttal into the final version and consolidate our response accordingly.

---

### Official Review · Reviewer_kC9o · 2025-07-02

**Clarity:** 3
**Significance:** 3
**Originality:** 3
**Rating:** 4
**Confidence:** 3

**Summary:**

This paper provides a systematic analysis of the phenomenon that as multimodal reasoning models enhance their reasoning abilities, their visual grounding capabilities deteriorate and hallucination issues become more pronounced. By comparing models of different scales and training paradigms, the authors show that longer reasoning chains make models increasingly rely on language priors at the cost of visual evidence, leading to higher hallucination rates. To address this, the paper proposes the RH-AUC metric and the RH-Bench diagnostic set to systematically quantify the trade-off between reasoning and hallucination, and provides an in-depth analysis of how various factors influence this balance.

**Questions:**

The conclusions drawn in this paper are meaningful, but the current experiments lack evaluation across more model series and sizes (see W2 & W3), raising concerns about the reliability and generalizability of the findings. Additionally, if the authors could further explore different RL methods and examine whether specific types of hallucination are linked to particular data or tasks (e.g., whether hallucinations are more common in counting scenarios), it would significantly enhance the value of this work (W4).

**Ethical Concerns:**

["NO or VERY MINOR ethics concerns only"]

**Final Justification:**

I appreciate the authors' efforts, and they have addressed my concerns during the rebuttal period.

**Limitations:**

yes

**Quality:**

3

**Strengths And Weaknesses:**

***Strengths:***
1. The paper introduces a novel metric (RH-AUC) and a comprehensive benchmark (RH-Bench) for systematically quantifying the trade-off between reasoning and hallucination in multimodal reasoning models.
2. Through extensive experiments and attention analysis, the study reveals and substantiates the important phenomenon that enhancing reasoning abilities in MLLMs can increase visual hallucinations, especially with longer reasoning chains.
3. The paper evaluates different training paradigms and model scales, demonstrating that the reasoning-hallucination trade-off is a general phenomenon, and provides useful insights for future model development.

***Weaknesses:***
1. The size of the RH-Bench benchmark is limited, which raises concerns about whether it is sufficient for a comprehensive evaluation of reasoning-hallucination trade-offs across diverse tasks and settings.
2. The experiments are limited to a single model series (mainly Qwen2.5-VL), with little evaluation on other mainstream multimodal model families.
3. The study does not include a sufficient range of model sizes and architectures, making it difficult to assess the generality of the conclusions across diverse model types.
4. The paper does not provide a more detailed analysis of hallucination types or investigate whether different categories of hallucination are related to specific data characteristics, which could potentially lead to a deeper understanding of the problem.

---

> ### Author Rebuttal · Authors · 2025-07-30
>
> We appreciate the reviewer's thoughtful comments and recognition of **our novel evaluation metric and benchmark, systematic analysis of reasoning-hallucination trade-offs, and comprehensive experiments across training paradigms**. We sincerely hope our response below can address your concerns.
>
> **(W1) Scale of RH-Bench and Effective Evaluation.**  Thank you for your insightful suggestion. The effectiveness of the evaluation provided by RH-Bench arises from its careful design, rather than solely relying on sample size. As illustrated in Figs. 10-11 of Appendix D, the non-monotonic trend in model performance with varying reasoning lengths observed in RH-Bench closely aligns with trends observed in datasets such as MathVista and MMhalu (Fig. 7 in manuscript), demonstrating the capability of the benchmark to reliably capture model behaviors. Furthermore, RH-Bench effectively distinguishes performance differences among models. This capability results from the design of task diversity, with reasoning tasks involving core domains such as math, visual common sense, and scientific questions, while perception tasks focus on detail recognition and object identification across both multiple-choice and open-ended formats. We plan to further expand the scale and dimensions of RH-Bench to facilitate more comprehensive and in-depth model evaluations.
>
> **(W2 & W3) Generalizability Verification Across More Model Families.**  Thank you for your valuable feedback. In response to W2 and W3 on model size and framework, most existing multimodal reasoning models often adopt Qwen2.5-VL as a backbone model and fine-tune on it. Therefore, the set of models we evaluated covers the same model series. To further demonstrate generalizability, we conduct additional experiments on recently released reasoning models and baselines with different model series: Qwen2-VL (7B), InternVL2.5 (8B, 38B), and Qwen2.5-VL (32B, 72B). These models are tested on four perception benchmarks and the RH-Bench. As shown in Table 1, the results from these broader models are consistent with our findings, providing strong evidence of the general applicability of our study. We will include the new results in the revision of the paper.
>
> **Table 1: Performance of recently released multimodal reasoning models on perception benchmarks and RH-Bench.**
> | Method             | Train Setting | Training Data | Training Data | MMVP     | MMEval-Pro | VMCBench | MMHALU | MMHALU | RH-Bench |
> |--------------------|---------------|---------------|---------------|----------|------------|----------|--------|--------|--------|
> |                    |               | Perc.         | Reas.         | Acc ↑    | Acc↑        | Acc↑      | Score↑ | Hallu↓ | RH-AUC↑    |
> | **Qwen2-VL-7B**        | Base          | -             | -             | 45.7     | 75.0       | 82.7     | 3.48   | 0.36   | -      |
> | R1-VL [1]          | SFT+RL        | 21k           | 40k           | 43.8     | 73.7       | 81.5     | 3.40   | 0.39   | 0.47   |
> | **InternVL2.5-8B**     | Base          | -             | -             | 45.0     | 74.6       | 81.5     | 3.52   | 0.35   | -      |
> | MM-Eureka [2]      | RL            | 19k           | 38k           | 44.2     | 72.5       | 80.0     | 3.48   | 0.38   | 0.53   |
> | **InternVL2.5-38B**    | Base          | -             | -             | 49.5     | 78.0       | 83.3     | 3.57   | 0.33   | -      |
> | MM-Eureka [2]      | RL            | -             | 9k            | 47.6     | 77.2       | 81.6     | 3.53   | 0.35   | 0.64   |
> | **Qwen2.5-VL-32B**     | Base          | -             | -             | 50.4     | 78.5       | 84.0     | 3.63   | 0.32   | -      |
> | MM-Eureka-Qwen [3] | RL            | -             | 17k           | 48.5     | 77.9       | 82.8     | 3.60   | 0.34   | 0.63   |
> | Vision-R1 [4]      | SFT+RL        | 60k           | 10k           | 47.8     | 77.0       | 81.9     | 3.56   | 0.36   | 0.60   |
> | **Qwen-2.5-VL-72B**    | Base          | -             | -             | 51.0     | 79.3       | 86.1     | 3.70   | 0.30   | -      |
> | VL-Rethinker [5]   | RL            | 6k            | 14k           | 49.7     | 77.8       | 84.9     | 3.67   | 0.33   | 0.68   |
> | Vision-R1 [4]      | SFT+RL        | 60k           | 10k           | 50.8     | 78.0       | 84.8     | 3.63   | 0.37   | 0.65   |
>
> **(W4) Analysis of Hallucination Across Seven Data Types.**  Thank you for your thoughtful feedback. We agree that performing this analysis could lead to a deeper understanding; therefore, following the suggestion, we conduct an analysis comparing the average hallucination rates of reasoning models to baseline models across 7 categories of hallucination. As shown in Table 2, the results indicate that most reasoning models exhibit higher hallucination rates across different hallucination types, aligning with our previous finding. Specifically, reasoning models demonstrate comparatively higher hallucination rates in tasks like existence, color, counting, OCR, and Chart & Doc, and the gap is larger on OCR and Chart & Doc, which might require more advanced visual capability.
>
> **Table2: Hallucination rates (%) of reasoning models across different hallucination types.**
> | Method                 | Presence | Count | Position | Color | OCR | Chart & Doc | Others |
> |------------------------|----------|-------|----------|-------|-----|-------------|--------|
> | Curr-ReFT-3B           | 30       | 35    | 23       | 12    | 45  | 36          | 25     |
> | Vision-R1-3B           | 32       | 33    | 25       | 9     | 43  | 40          | 24     |
> | **Qwen2.5-VL-3B**          | 28       | 34    | 17       | 12    | 43  | 34          | 23     |
> | R1-OneVision-7B        | 28       | 31    | 20       | 5     | 38  | 33          | 22     |
> | ThinkLite-VL-7B        | 20       | 32    | 19       | 10    | 36  | 30          | 20     |
> | **Qwen2.5-VL-7B**         | 23       | 30    | 15       | 7     | 30  | 27          | 18     |
> | MM-Eureka-Qwen-32B     | 19       | 26    | 13       | 6     | 29  | 25          | 15     |
> | **Qwen2.5-VL-32B**         | 18       | 23    | 10       | 5     | 26  | 22          | 10     |
> | VL-Rethinker-72B       | 19       | 22    | 13       | 6     | 28  | 20          | 12     |
> | **Qwen2.5-VL-72B**         | 16       | 20    | 8        | 5     | 25  | 19          | 9      |
>
>
>
> [1] *R1-VL: Learning to Reason with Multimodal Large Language Models via Step-wise Group Relative Policy Optimization*
>
> [2] *MM-Eureka: Exploring visual aha moment with rule-based large-scale reinforcement learning*
>
> [3] *MM-Eureka: Exploring the frontiers of multimodal reasoning with rule-based reinforcement learning*
>
> [4] *Vision-R1: Incentivizing Reasoning Capability in Multimodal Large Language Models*
>
> [5] *VL-Rethinker: Incentivizing Self-Reflection of Vision-Language Models with Reinforcement Learning*

---

> > ### Author Response · Authors · 2025-08-06
> >
> > Dear Reviewer kC9o,
> >
> > We greatly appreciate your feedback, which has been invaluable in improving our work. We have carefully addressed your comments about sufficiency of RH-Bench evaluation, generalizability verification across multiple model families, and analysis of hallucination types.
> >
> > As the discussion period is ending in two days, we kindly invite you to review our response. We hope our response addresses adequately your concerns. Thanks!
> >
> > Best,
> >
> > The Authors

---

> > > ### Comment · Reviewer_kC9o · 2025-08-07
> > >
> > > Thank you for your response.
> > >
> > > First, I acknowledge that this work is both meaningful and inspiring. However, from the perspective of paper acceptance, I am mainly concerned about the generalization and the relatively high cost of exploring more models.
> > >
> > > Therefore, I will ultimately consider either maintaining the original score or raising it to 4.

---

> ### Author Response · Authors · 2025-08-08
>
> Thank you for your continued engagement and for raising these important questions. We sincerely appreciate the dialogue, as your feedback is helping us to sharpen the positioning and contribution of our work. We would like to address your remaining concerns.
>
> Regarding the concerns about exploring a wider range of model costs, we would like to point out that the cost of evaluating a reasoning model's RH-AUC on our proposed RH-Bench is similar to performing inference of the model on the same benchmark dataset, as our method does not require extra training of the model, and different length of reasoning are independant and therefore can be parallelarized. As shown in Table 1, we compare the computation time of RH-AUC (with different length-control methods) with the total inference time of the original model on RH-Bench. The results clearly show that the average computation time of our steering method (LSS) for RH-AUC differs only marginally from the inference time of the original model on RH-Bench,  while demonstrating greater efficiency compared to the other two control methods.
>
> **Table1:** Comparison of inference time (s) under various reasoning control methods on the RH-Bench for RH-AUC evaluation.  Three reasoning length control methods: Token Budget Forcing (Budget), Test Time Scaling (Scaling), Latent State Steering (LSS).  Original refers to the model's normal reasoning.
>
> | **Method**                | Original ↓ | Budget ↓ | Scaling ↓ | LSS ↓ |
> |---------------------------|------------|----------|-----------|-------|
> | **Qwen2.5-VL-3B**         | -          | -        | -         | -     |
> | LLM-R1-3B                  | 728        | 735      | 740       | **730** |
> | Curr-ReFT-3B               | 807        | 816      | 824       | **806** |
> | Ocean-R1-3B                | 854        | 859      | 863       | **856** |
> | **Qwen2-VL-7B**           | -          | -        | -         | -     |
> | R1-VL-7B                   | 905        | 907      | 914       | **906** |
> | **Qwen2.5-VL-7B**         | -          | -        | -         | -     |
> | R1-OneVision-7B            | 955        | 963      | 965       | **957** |
> | ThinkLite-VL-7B            | 1003       | 1007     | 1012      | **1003** |
> | OpenVLThinker-7B           | 1152       | 1158     | 1160      | **1150** |
> | MM-Eureka-7B               | 972        | 978      | 980       | **974** |
> | **InternVL2.5-8B**        | -          | -        | -         | -     |
> | MM-Eureka-8B               | 1009       | 1016     | 1020      | **1012** |
> | **InternVL2.5-38B**       | -          | -        | -         | -     |
> | MM-Eureka-38B              | 1107       | 1109     | 1108      | **1107** |
> | **Qwen2.5-VL-32B**        | -          | -        | -         | -     |
> | MM-Eureka-Qwen-32B         | 1195       | 1200     | 1203      | **1198** |
> | Vision-R-32B               | 1226       | 1231     | 1234      | **1227** |
> | **Qwen-2.5-VL-72B**       | -          | -        | -         | -     |
> | VL-Rethinker-72B           | 1275       | 1279     | 1281      | **1274** |
> | Vision-R1-72B              | 1288       | 1295     | 1298      | **1290** |

---

> ### Author Response · Authors · 2025-08-08
>
> Regarding generalization, our comprehensive evaluation of existing multimodal reasoning models is highly consistent with the findings in this paper. Specifically, as shown in Table 2, models trained with RL-only methods, including the recently proposed VL-Rethinker and Vision-R1, generally outperform SFT+RL trained models on the key metric RH-AUC. This suggests that RL-only training may be more effective at optimizing the reasoning-perception balance. Furthermore, as shown in Table 3, the recently released MM-Eureka-Qwen surpasses Vision-R1 (trained on 70k samples) despite using a smaller dataset focused on reasoning data. This indicates that simply increasing the training data scale does not guarantee a better reasoning-perception balance. The results in Table 4 consistently show a positive correlation between model size and the reasoning-perception balance. Finally, as indicated in Table 5, all current multimodal reasoning models suffer from an amplified hallucination problem in perception tasks.
>
> We believe these new results provide strong empirical evidence that our framework is efficient and generalizable. Thank you again for your time and valuable insights. We hope that this new body of evidence and our detailed clarifications will be helpful in your final assessment of our work. **We would be very grateful if you would consider these new results in your final evaluation.**
>
> **Table2:** Comparison of RL and SFT+RL training paradigms on RH-AUC.
>
> | **Method**                | **Train Setting** | **RH-AUC ↑** |
> |---------------------------|-------------------|--------------|
> | **Qwen2-VL-7B**               | **Base**              | -            |
> | R1-VL                     | SFT+RL            | 0.47         |
> | **Qwen2.5-VL-3B**             | **Base**              | -            |
> | LLM-R1-3B                 | RL                | 0.46         |
> | Curr-ReFT-3B              | SFT+RL            | 0.47         |
> | Ocean-R1-3B               | RL                | 0.53         |
> | **Qwen2.5-VL-7B**             | **Base**              | -            |
> | R1-OneVision-7B           | SFT+RL            | 0.46         |
> | ThinkLite-VL-7B           | RL                | 0.52         |
> | OpenVLThinker-7B          | SFT+RL            | 0.54         |
> | MM-Eureka-7B              | RL                | 0.55         |
> | MM-R1-7B                  | RL                | 0.57         |
> | Ocean-R1-7B               | RL                | 0.63         |
> | **InternVL2.5-38B**           | **Base**              | -            |
> | MM-Eureka-38B             | RL                | 0.64         |
> | **Qwen2.5-VL-32B**            | **Base**              | -            |
> | MM-Eureka-Qwen-32B        | RL                | 0.63         |
> | Vision-R1-32B             | SFT+RL            | 0.60         |
> | **Qwen2.5-VL-72B**            | **Base**              | -            |
> | VL-Rethinker-72B          | RL                | 0.68         |
> | Vision-R1-72B             | SFT+RL            | 0.65         |

---

> ### Author Response · Authors · 2025-08-08
>
> **Table3:** Comparison of RH-AUC under diverse training data scales and types.
>
> | **Method**                | **Perc. Data** | **Reas. Data** | **RH-AUC ↑** |
> |---------------------------|----------------|----------------|--------------|
> | **Qwen2-VL-7B**               | -              | -              | -            |
> | R1-VL                     | 21k            | 40k            | 0.47         |
> | **Qwen2.5-VL-3B**             | -              | -              | -            |
> | LLM-R1-3B                 | 65k            | 40k            | 0.46         |
> | Curr-ReFT-3B              | 6k             | 3k             | 0.47         |
> | Ocean-R1-3B               | 20k            | 63k            | 0.53         |
> | **Qwen2.5-VL-7B**             | -              | -              | -            |
> | R1-OneVision-7B           | 80k            | 77k            | 0.46         |
> | ThinkLite-VL-7B           | 62k            | 8k             | 0.52         |
> | OpenVLThinker-7B          | 25k            | 25k            | 0.54         |
> | MM-Eureka-7B              | -              | 15k            | 0.55         |
> | MM-R1-7B                  | -              | 6k             | 0.57         |
> | Ocean-R1-7B               | 20k            | 63k            | 0.63         |
> | **InternVL2.5-8B**            | -              | -              | -            |
> | MM-Eureka-8B              | 19k            | 38k            | 0.53         |
> | **InternVL2.5-38B**           | -              | -              | -            |
> | MM-Eureka-38B             | -              | 9k             | 0.64         |
> | **Qwen2.5-VL-32B**            | -              | -              | -            |
> | MM-Eureka-Qwen-32B        | -              | 17k            | 0.63         |
> | Vision-R1-32B             | 60k            | 10k            | 0.60         |
> | **Qwen2.5-VL-72B**            | -              | -              | -            |
> | VL-Rethinker-72B          | 6k             | 14k            | 0.68         |
> | Vision-R1-72B             | 60k            | 10k            | 0.65         |
>
> **Table4:** Comparison of RH-AUC across model sizes including 3B, 7B, 8B, 32B, 38B, and 72B.
>
> | **Method**                | **Size** | **RH-AUC ↑** |
> |---------------------------|----------|--------------|
> | **Qwen2.5-VL**             | **3B**       | -            |
> | LLM-R1-3B                 | 3B       | 0.46         |
> | Curr-ReFT-3B              | 3B       | 0.47         |
> | Ocean-R1-3B               | 3B       | 0.53         |
> | **Qwen2.5-VL**             | **7B**       | -            |
> | R1-OneVision-7B           | 7B       | 0.46         |
> | ThinkLite-VL-7B           | 7B       | 0.52         |
> | OpenVLThinker-7B          | 7B       | 0.54         |
> | MM-Eureka-7B              | 7B       | 0.55         |
> | MM-R1-7B                  | 7B       | 0.57         |
> | Ocean-R1-7B               | 7B       | 0.63         |
> | **Qwen2.5-VL**            | **32B**      | -            |
> | Vision-R1-32B             | 32B      | 0.60         |
> | MM-Eureka-Qwen-32B        | 32B      | 0.63         |
> | **InternVL2.5**           | **38B**      | -            |
> | MM-Eureka-38B             | 38B      | 0.64         |
> | **Qwen2.5-VL**            | **72B**      | -            |
> | VL-Rethinker-72B          | 72B      | 0.68         |
> | Vision-R1-72B             | 72B      | 0.65         |

---

> ### Author Response · Authors · 2025-08-08
>
> **Table5:** Comparison of reasoning and non-reasoning models on four perception benchmarks.
>
> | **Method**                | **MMVP ↑** | **MMEval-Pro ↑** | **VMCBench ↑** | **MMHALU Score ↑** | **MMHALU Hallu ↓** |
> |---------------------------|------------|------------------|----------------|--------------------|--------------------|
> | **Qwen2-VL-7B**           | **45.7**   | **75.0**         | **82.7**       | **3.48**           | **0.36**           |
> | R1-VL                     | 43.8       | 73.7             | 81.5           | 3.40               | 0.39               |
> | **Qwen2.5-VL-3B**         | **36.0**   | **67.8**         | **77.6**       | **3.33**           | **0.43**           |
> | LLM-R1-3B                 | 24.0       | 66.9             | 76.0           | 2.27               | 0.44               |
> | Curr-ReFT-3B              | 28.7       | 60.2             | 73.0           | 2.78               | 0.45               |
> | Ocean-R1-3B               | 39.3       | 62.5             | 76.5           | 2.35               | 0.46               |
> | **Qwen2.5-VL-7B**         | **47.3**   | **76.0**         | **84.5**       | **3.50**           | **0.33**           |
> | R1-OneVision-7B           | 43.0       | 69.4             | 65.2           | 3.20               | 0.48               |
> | ThinkLite-VL-7B           | 47.0       | 72.0             | 83.4           | 3.30               | 0.38               |
> | OpenVLThinker-7B          | 46.5       | 71.5             | 80.5           | 3.00               | 0.36               |
> | MM-Eureka-7B              | 46.7       | 74.8             | 82.0           | 3.20               | 0.43               |
> | MM-R1-7B                  | 44.3       | 73.6             | 80.3           | 3.10               | 0.38               |
> | Ocean-R1-7B               | 52.3       | 73.7             | 82.3           | 2.80               | 0.35               |
> | **InternVL2.5-8B**        | **45.0**   | **74.6**         | **81.5**       | **3.52**           | **0.35**           |
> | MM-Eureka-8B              | 44.2       | 72.5             | 80.0           | 3.48               | 0.38               |
> | **InternVL2.5-38B**       | **49.5**   | **78.0**         | **83.3**       | **3.57**           | **0.33**           |
> | MM-Eureka-38B             | 47.6       | 77.2             | 81.6           | 3.53               | 0.35               |
> | **Qwen2.5-VL-32B**        | **50.4**   | **78.5**         | **84.0**       | **3.63**           | **0.32**           |
> | MM-Eureka-Qwen-32B        | 48.5       | 77.9             | 82.8           | 3.60               | 0.34               |
> | Vision-R1-32B             | 47.8       | 77.0             | 81.9           | 3.56               | 0.36               |
> | **Qwen2.5-VL-72B**        | **51.0**   | **79.3**         | **86.1**       | **3.70**           | **0.30**           |
> | VL-Rethinker-72B          | 49.7       | 77.8             | 84.9           | 3.67               | 0.33               |
> | Vision-R1-72B             | 50.8       | 78.0             | 84.8           | 3.63               | 0.37               |

---

### Author Response · Authors · 2025-08-09

For clarity and simplicity, we will refer to Reviewers kC9o, u6Et, cecd, and qZ13 as R1, R2, R3, and R4, respectively, in the following response.

We sincerely thank all reviewers for their thoughtful and constructive feedback. We are encouraged by their recognition of the key contributions and strengths of our work.

In particular, we thank the reviewers for highlighting the importance and timeliness of the hallucination amplification phenomenon in   multimodal reasoning and the perception-reasoning balance problem addressed in this work (**R1, R3, R4**). We also appreciate their recognition of our proposed RH-AUC metric and RH-Bench comprehensive benchmark, which systematically quantify the trade-off between reasoning and hallucination in multimodal reasoning models (**R1, R2, R3, R4**). Moreover, we are grateful for their acknowledgement of our large-scale experiments and attention analyses that reveal and validate the phenomenon of hallucination amplification in multimodal reasoning models (**R1, R3, R4**). Additionally, we are pleased to see reviewers endorse our analytical finding that the reasoning chain length exhibits a non-monotonic relationship with reasoning–perception performance, with the optimal length varying by task (**R3, R4**). They also affirm that our systematic comparisons across different model scales and training paradigms provide valuable insights for future model development (**R1, R3, R4**). Finally, we are encouraged that reviewers unanimously find the paper well-structured, logically coherent, and easy to follow (**R1, R2, R3, R4**).

We look forward to the reviewers' favorable consideration and remain grateful for their valuable feedback.

---

### Note · Authors · 2025-08-13

We sincerely thank all the reviewers for their valuable insights and for recognizing the key strengths of our work.

1. Importance and timeliness of hallucination amplification in multimodal reasoning , with recognition of large-scale experiments and attention analyses validating this phenomenon.
2. Analytical finding of a non-monotonic relationship between reasoning chain length and reasoning–perception performance, with optimal length varying by task.
3. RH-AUC metric and RH-Bench benchmark as systematic tools for quantifying the trade-off between reasoning and hallucination in multimodal reasoning models.
4. Systematic comparisons across different model scales and training paradigms, providing valuable insights for future model development.

We will ensure that all additional experiments and clarifications from the rebuttal are included in the final version.

1. A major concern shared by multiple reviewers is the generalizability of our findings. We extended our experiments beyond Qwen2.5-VL to include multiple model series and parameter scales. Results consistently confirm that all current multimodal reasoning models exhibit amplified hallucination in perception tasks.
2. To further investigate the differences in reasoning behavior between RL and SFT+RL models, we compared the average response token length and the frequency of reasoning-related vocabulary in the outputs of the two paradigms. SFT+RL models produced significantly more redundant responses than RL models, which is consistent with our findings. In the revised version, we will further clarify that our current analysis, grounded in systematic observations of the performance of off-the-shelf models.

Finally, we sincerely thank the reviewer for the thoughtful feedback and for carefully engaging with our rebuttal. Our work is the first, to our knowledge, to uncover and systematically analyze the phenomenon of hallucination amplification in multimodal reasoning models. This issue, where increasing reasoning length exacerbates hallucinations, has not been previously studied in the literature, and we believe it addresses an urgent and underexplored challenge in the development of reliable vision-language systems. To study this issue, we introduce a comprehensive evaluation framework, including RH-AUC and RH-Bench, to precisely measure the interaction between reasoning and visual grounding, thereby providing actionable insights into the reasoning–perception trade-off.

---

### Decision · Program_Chairs · 2025-09-17

**Decision:**

Accept (poster)

**Comment:**

This paper analyzes how longer reasoning chains in multimodal models amplify hallucinations by weakening visual grounding. It introduces the RH-AUC metric and RH-Bench benchmark to quantify this trade-off, and the experiments provide evidence and insights. While questions remain about benchmark scale and novelty as pointed out by R, the rebuttal addressed them with some broader results. Overall, this is a timely and valuable contribution AC thinks. AC recommend weak accept.